METHODS AND RESOURCES

# Multivariate phenotype analysis enables genome-wide inference of mammalian gene function

**George Nicholson**[1,2]*, **Hugh Morgan**[2], **Habib Ganjgahi**[1,2], **Steve D. M. Brown**[2], **Ann-Marie Mallon**[2], **Chris Holmes**[1,2]

**1** University of Oxford, Oxford, United Kingdom, **2** MRC Harwell Institute, Harwell, United Kingdom

* george.nicholson@stats.ox.ac.uk

**Data Availability Statement:** The authors confirm that all data underlying the findings are fully available without restriction. The data and code used to generate the results in the paper are

## Abstract

The function of the majority of genes in the human and mouse genomes is unknown. Investigating and illuminating this dark genome is a major challenge for the biomedical sciences. The International Mouse Phenotyping Consortium (IMPC) is addressing this through the generation and broad-based phenotyping of a knockout (KO) mouse line for every protein-coding gene, producing a multidimensional data set that underlies a genome-wide annotation map from genes to phenotypes. Here, we develop a multivariate (MV) statistical approach and apply it to IMPC data comprising 148 phenotypes measured across 4,548 KO lines.

There are 4,256 (1.4% of 302,997 observed data measurements) hits called by the univariate (UV) model analysing each phenotype separately, compared to 31,843 (10.5%) hits in the observed data results of the MV model, corresponding to an estimated 7.5-fold increase in power of the MV model relative to the UV model. One key property of the data set is its 55.0% rate of missingness, resulting from quality control filters and incomplete measurement of some KO lines. This raises the question of whether it is possible to infer perturbations at phenotype–gene pairs at which data are not available, i.e., to infer some in vivo effects using statistical analysis rather than experimentation. We demonstrate that, even at missing phenotypes, the MV model can detect perturbations with power comparable to the single-phenotype analysis, thereby filling in the complete gene–phenotype map with good sensitivity.

A factor analysis of the MV model's fitted covariance structure identifies 20 clusters of phenotypes, with each cluster tending to be perturbed collectively. These factors cumulatively explain 75% of the KO-induced variation in the data and facilitate biological interpretation of perturbations. We also demonstrate that the MV approach strengthens the correspondence between IMPC phenotypes and existing gene annotation databases. Analysis of a subset of KO lines measured in replicate across multiple laboratories confirms that the MV model increases power with high replicability.

available at https://github.com/georgenicholson/multivariate_phenotype_data_and_code and https://zenodo.org/record/6787112.

**Funding:** This work was supported by Medical Research Council (https://mrc.ukri.org/) Programme grant MC_UP_A390_1107 (G.N., H.G. and C.H.) and National Institutes of Health (https://www.nih.gov/) grant U54 HG006370 (H.M., S.D.M.B., A.-M.M.). The funders had no role in study design, data collection and analysis, decision to publish, or preparation of the manuscript.

**Competing interests:** The authors have declared that no competing interests exist.

**Abbreviations:** BP, Biological Process; EM, expectation–maximisation; eQTL, expression quantitative trait loci; Fdr, false discovery rate; Fsr, false sign rate; GO, Gene Ontology; IMPC, International Mouse Phenotyping Consortium; KO, knockout; LOO-MV, leave-one-procedure-out MV; MAP, maximum a posteriori; MAR, missing at random; mash, multivariate adaptive shrinkage; MCMC, Markov chain Monte Carlo; MV, multivariate; OR, odds ratio; UV, univariate; WT, wild-type; XD, Extreme Deconvolution.

# Introduction

The function of the majority of genes in the human and mouse genomes is unknown. Investigating and illuminating this dark genome is a major challenge for the biomedical sciences [1]. Developing a comprehensive catalogue of mammalian gene function will be a vital underpinning to studies of rare and common disease and advances in precision medicine. The International Mouse Phenotyping Consortium (IMPC) is a collaboration between 21 research institutions worldwide aimed at addressing the challenge of the dark genome through the generation and broad-based phenotyping of a knockout (KO) mouse line for every protein-coding gene (www.mousephenotype.org).

In excess of 300 measurements are conducted on each animal, ranging from clinical blood chemistry, through calorimetry and body composition, to behavioural phenotypes [2]. By inference from the multidimensional data sets produced, the IMPC is compiling a genome-wide annotation map from genes to phenotypes that is already providing unique insights into mammalian gene function and the genome landscape of diverse diseases [3,4,5,6,7,8].

By March 2022, approximately 10,000 KO mouse lines, many for poorly understood genes, have so far been generated, and 8,623 of those lines have been phenotyped using standardised procedures for a wide variety of disease systems. In this paper, we analyse a partial IMPC data set comprising 4,548 KO lines with phenotype data from some of 148 quantitative phenotypes as of 26 March 2018.

In the IMPC adult phenotyping pipeline, a sequence of standardised measurements is performed on single-gene KO and control mice aged between 9 and 16 weeks. We refer to the measurements as *phenotypes* with these being measured in groups called *procedures*; all phenotypes within a given procedure are measured in a specific week of age (S1 Fig). The scientific purpose, experimental design, and detailed description for each procedure are presented at the IMPC website [9]. The primary scientific goal is to identify statistically significant KO-induced phenotypic perturbations, also referred to as phenotypic *hits* or *positive annotations*. The experimental design of the IMPC measures on average 14 animals (7 of each sex) from each KO line, contemporaneously with the rolling baseline of control animals. This is visualised for a pair of phenotypes at one of the phenotyping centres, MRC Harwell, in Fig 1. The statistical goal is to estimate and test for a difference in phenotypic mean between each KO line and the shared set of control animals. Conceptually, an unpaired *t* test between KOs and controls is the basic statistical idea, but in practice, multilevel modelling is necessary due to the complex experimental structure. For example, litters and other experimental strata are occasionally confounded with the gene-KO effects of interest, necessitating the use of hierarchical models to identify effects of interest [10,11,12].

So far, the literature on high-throughput phenotyping has focused exclusively on calling hits (testing for a perturbation) at one phenotype at a time, using so-called univariate (UV) models [11,12]. However, initial results from the IMPC have revealed strong correlation between perturbations at different phenotypes. Multivariate (MV) association methods have already proven successful in many genetic applications, such as genome-wide association studies [15,16,17] and multi-tissue eQTL studies [18,19,20,21,22]. This points to an opportunity for improving inference in the IMPC by sharing information across phenotypes using MV methods. In particular, when sample size is severely limited on ethical and financial grounds, the hope is that MV methods can computationally increase the information extracted from the data that are gathered. Further, in our IMPC data set, not all phenotypes are available on each KO line. This raises the question of whether it is possible to infer perturbations at (phenotype, KO line) pairs at which data are not available, i.e., to infer some in vivo effects using statistical analysis rather than experimentation. We set out to implement an MV model that can effectively perform this type of inference when some data are missing.

(a) Triglycerides, phenotype p     (b) Body fat percentage, phenotype $\tilde{p}$

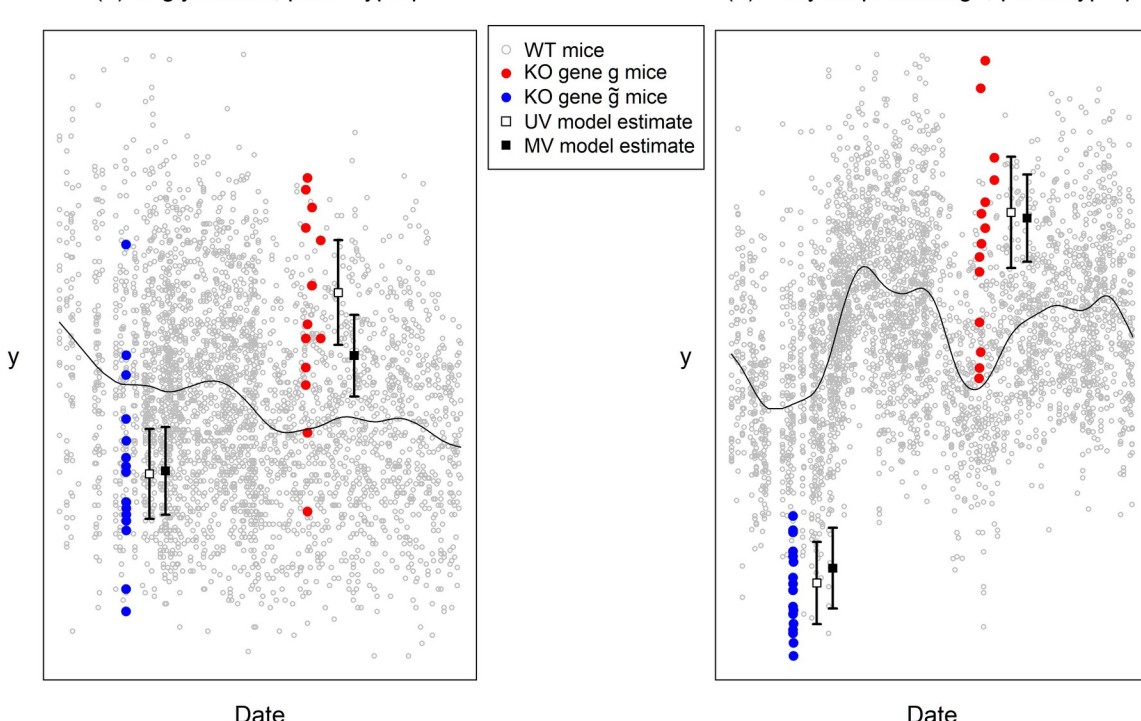

**Fig 1. Experimental design of the IMPC.** Each point corresponds to 1 animal, with data from 2 KO lines—labelled $g$ and $\tilde{g}$—displayed alongside contemporaneous data from a large number of control (wild-type, or WT) animals in grey (see legend). Panels (a) and (b) show data from phenotypes $p$ (Triglycerides) and $\tilde{p}$ (Body fat percentage), respectively. Our goal is to quantify the underlying expected perturbations of the red/blue coloured points from the rolling WT baseline (illustrated with a smooth black curve), in the presence of structured experimental noise. Annotated on the plot, to the right of the red/blue measurement data of each gene–phenotype pair, are the posterior mean estimates from the UV and MV models, $\widehat{\theta}_{pg}^{\mathrm{UV}}$ with empty squares and $\widehat{\theta}_{pg}^{\mathrm{MV}}$ with filled squares (see legend), along with error bars denoting ±2 posterior SDs. In the current paper, we combine the information in $\widehat{\theta}_{pg}^{\mathrm{UV}}$ across multiple related phenotypes, such as $p$ and $\tilde{p}$, thereby generating improved estimators $\widehat{\theta}_{pg}^{\mathrm{MV}}$. The relative means and SDs of the UV and MV estimators shown in the plot are illustrative of their general properties—MV posterior means are shrunken towards zero (towards the black curve here) relative to the UV posterior means in 90.2% of cases (phenotype–gene pairs); while MV posterior SDs are smaller than UV posterior SDs in > 99.9% of cases. The data and code used to generate this figure are available at [13,14]. IMPC, International Mouse Phenotyping Consortium; KO, knockout; MV, multivariate; UV, univariate; WT, wild-type.

We adopt a composable approach to MV modelling that is computationally attractive while effectively capturing the important variation in the IMPC data set. First, we fit a UV multilevel model [11] for each phenotype separately. Second, we take the UV model's outputted effect estimates and standard errors and fit an MV model to these, building methodologically on the work of [22,23]. We contextualise and compare performance of our methods against the background of this existing work in Methods–*Comparison with existing methods*.

A major goal of the IMPC is to create a comprehensive gene–phenotype annotation map. From a statistical perspective, this involves testing the null hypothesis that there is no phenotypic perturbation. Alongside the MV model, we design a permutation-based approach to hypothesis testing aimed at powerful inference under careful control and monitoring of false positive rates. Our approach is based on the generation of synthetic null KO lines by structured random resampling from control animals (details in Methods–*Control of error rates*). By analysing synthetic null lines alongside true KO lines, we are able to select significance thresholds for effective error rate control.

We validate our MV method in a number of ways. We evaluate the efficacy of inference in the presence of missing data by artificially masking data and comparing the masked data results to the fully observed data results. We independently assess the MV hit-calling method by examining the replicability of hits called on the same KO lines measured across multiple laboratories. We also perform a number of additional checks, around the biological reasonableness of the results, as well as assessing quantitative measures of model robustness and fit. Our checks indicate that the MV approach can substantially increase hit rates in the IMPC, while retaining error rate control and replicability, even when calling hits in cases of missing phenotype data. The development of a sensitive, replicable, and comprehensive gene–phenotype map will ensure that the number of animals used in follow-up experiments to the IMPC is minimised, in alignment with the 3Rs of replacement, reduction, and refinement [24].

## Results

We have previously designed a UV Bayes linear multilevel model targeting the phenotypic perturbation of gene KO animals relative to wild-type (WT) animals [11]. We fit this model to each (phenotype, centre) combination separately, yielding an estimate (and SE) of the phenotypic perturbation, $\widehat{\theta}_{pg}^{\text{UV}}$ (and $\widehat{s}_{pg}^{\text{UV}}$), for each (phenotype $p = 1,\ldots,P$, gene $g = 1,\ldots,G$) pair at which measurements are available. Example data and estimates of $\widehat{\theta}_{pg}^{\text{UV}}$ are illustrated in Fig 1.

In this paper, we develop an MV modelling framework, building on the methodological work of [22,23]. The method takes as input the UV results, $\widehat{\theta}_{pg}^{\text{UV}}$ ($\widehat{s}_{pg}^{\text{UV}}$), and outputs MV estimates $\widehat{\theta}_{pg}^{\text{MV}}$ ($\widehat{s}_{pg}^{\text{MV}}$) across all (phenotype $p$, gene $g$) combinations, including those pairs at which data are unavailable. The MV model is based on a covariance structure $\Sigma$ allowing perturbations to be correlated across different phenotypes, as illustrated between Triglycerides and Body fat percentage in Fig 1. The method also incorporates a correlation matrix, $R$, to account for structure in experimental noise across phenotypes. A practically useful property of this 2-stage model is its *composability*, whereby results can be transferred efficiently between 2 different analyses or computational tools—here from an arbitrarily complex UV model to a highly structured MV model.

We lay out the results in 3 conceptual stages. First, we provide high-level technical descriptions of the UV and MV models. Second, we characterise the IMPC hit calling results, contrasting the UV and MV models, and with a focus on demonstrating statistical power and replicability. Finally, we look to applications to demonstrate how the MV approach can illuminate relationships between phenotypic perturbations and underlying biological mechanisms, and do this relatively effectively compared to its UV counterpart. These examples provide extra evidence of the MV method's validity and replicability by illustrating how its results make intuitive sense and are aligned with existing scientific knowledge.

### Univariate model

The parameter of interest throughout is denoted by $\theta_{pg}$ and represents the expected perturbation of the $p$th phenotype in the $g$th gene KO, relative to WT animals (Fig 1). This UV model, fitted only to data from KO line $g$ accompanied by data from the entire rolling baseline of WT animals, takes the form of a linear multilevel model (or mixed-effects model):

$$y_i = \theta_{pg}\mathbb{I}(\text{animal } i \text{ is in line } g) + \boldsymbol{x}_i^T\boldsymbol{\beta} + \sum_r \boldsymbol{z}_{ri}^T\boldsymbol{\alpha}_r + \varepsilon_i \tag{1}$$

$$\boldsymbol{\alpha}_r \sim \text{MVNormal}(0, \sigma_r^2\boldsymbol{I})$$

$$\boldsymbol{\varepsilon} \sim \mathrm{MVNormal}(0, \sigma_{\mathrm{resid}}^2 \boldsymbol{I}),$$

where $y_i$ is the Box–Cox transformed [25] measurement of the $p$th phenotype on the $i$th mouse. The parameters in $\boldsymbol{\beta}$ adjust additively for sex, sex–genotype interaction, strain, investigator, and other experimental metadata, while day and litter effects are modelled hierarchically via $\boldsymbol{\alpha}_{\mathrm{day}}$ and $\boldsymbol{\alpha}_{\mathrm{litter}}$ with variance components $\sigma_{\mathrm{day}}^2$ and $\sigma_{\mathrm{litter}}^2$. In this paper, we focus on estimation of $\theta_{pg}$, the main effect of genotype $g$ on phenotype $p$. In cases where genotype effects differ between sexes [5], $\theta_{pg}$ is interpretable as the average of those sex-specific effects. Longitudinal changes in the measurement baseline are modelled using a penalised spline which features in both fixed and random components [26]. Noninformative priors are specified for $\theta_{pg}$, $\boldsymbol{\beta}$ and the $\sigma_r$, with the model being fitted via Markov chain Monte Carlo (MCMC) and outputting samples from the marginal posterior distribution $p(\theta_{pg}|\boldsymbol{y})$ (for further details, see S1 Note and [11]).

The UV inference outputs an estimate and standard error for each $\theta_{pg}$, i.e., the posterior mean $\widehat{\theta}_{pg}^{\mathrm{UV}}$ and posterior SD $\widehat{s}_{pg}^{\mathrm{UV}}$, respectively. We perform careful quality control of the UV results, conservatively filtering out (from downstream MV analysis) any centre–procedure combinations that exhibit anomalous longitudinal patterns in UV results; such patterns can be indicative of unmodelled experimental artefacts rather than the biological effects (S2 Fig). Next, to ensure that there are sufficient data at each phenotype, we apply a post-QC heuristic filter whereby we retain only those phenotypes with UV effect estimates for at least 500 KO lines. After QC and filtering, the UV estimates (and SEs) are scaled so that the $\widehat{\theta}_{pg}^{\mathrm{UV}}$ have unit SD for each phenotype within each phenotyping centre and are then taken forward as input for the MV model.

## Multivariate model

In collecting together the results of the UV multilevel model, we obtain unbiased estimates $\widehat{\boldsymbol{\theta}}_{\cdot g}^{\mathrm{UV}}$ (and SEs $\widehat{s}_{pg}^{\mathrm{UV}}$) for $\boldsymbol{\theta}_{\cdot g}$ that are affected by MV experimental noise, having the covariance structure $\widehat{\boldsymbol{S}}_g^{\mathrm{UV}} \boldsymbol{R} \widehat{\boldsymbol{S}}_g^{\mathrm{UV}}$. Further, the latent $P$-dimensional MV perturbations $\boldsymbol{\theta}_{\cdot g}$ tend to exhibit strong $P{\times}P$ covariance structure. These aspects of the data suggest a model following the form of [22,23]:

$$\widehat{\boldsymbol{\theta}}_{\cdot g}^{\mathrm{UV}} = \mathrm{N}(\widehat{\boldsymbol{\theta}}_{\cdot g}^{\mathrm{UV}}|\boldsymbol{\theta}_{\cdot g}, \widehat{\boldsymbol{S}}_g^{\mathrm{UV}} \boldsymbol{R} \widehat{\boldsymbol{S}}_g^{\mathrm{UV}}) \tag{2}$$

$$p(\boldsymbol{\theta}_{\cdot g}|\boldsymbol{\Sigma}_{1:S}, \boldsymbol{\pi}) = \sum_{m=1}^{M} \sum_{s=1}^{S} \pi_{ms} \mathrm{N}(\boldsymbol{\theta}_{\cdot g}|\boldsymbol{0}, \omega_m \boldsymbol{\Sigma}_s) \tag{3}$$

where the parameters $\boldsymbol{\Sigma}_s$ represent the covariance of $\boldsymbol{\theta}_{\cdot g}$, i.e., of the expected phenotypic perturbation for a KO line, and the hyperparameter $\boldsymbol{R}$ models the correlation structure of the experimental noise. The $\widehat{\boldsymbol{S}}_g^{\mathrm{UV}} := diag\left(\widehat{s}_{1g}^{\mathrm{UV}}, \ldots, \widehat{s}_{Pg}^{\mathrm{UV}}\right)$ are known diagonal matrices of standard errors outputted by the UV model. The density of the latent perturbations, $p(\boldsymbol{\theta}_{\cdot g}|\boldsymbol{\Sigma}, \pi_{1:M})$, is an MV Gaussian mixture model with mixing probabilities $\pi_{1:M;1:S}$ over a specified ladder of scales given by constants $\omega_{1:M}$ [22,27], and $S{\geq}1$ covariance matrices $\boldsymbol{\Sigma}_{1:S}$ to be learned [23]. We relate our approach to [22,23] in more detail and compare performance in Methods–*Comparison with existing methods*.

We constrain $\boldsymbol{\Sigma}_s$ to factor-model form (see, e.g., [28]):

$$\boldsymbol{\Sigma}_s = \boldsymbol{W}_s \boldsymbol{W}_s^T + \boldsymbol{\Psi}_s, \tag{4}$$

where $W_s$ is a $P{\times}K$ matrix, and $\Psi_s$ a diagonal $P{\times}P$ matrix having positive diagonal elements. We performed the full analysis for fixed $K\in\{15,20,30,40\}$. The results presented in the manuscript are for a choice of $K = 20$ and $S = 1$, selected with reference to false discovery rate–controlled (Fdr-controlled) hit rates [29,30], and cross-validated likelihood measures of model fit (see Methods–*Comparison with existing methods*).

We take an empirical Bayes approach to inference in the MV model specified at (2)–(4). The experimental correlation hyperparameter, $R$, is estimated from synthetic null data and fixed at $\widehat{R}$ in advance [22]. The expectation–maximisation (EM) algorithm is used to obtain maximum a posteriori (MAP) estimates of hyperparameters $\Sigma_{1:S}$ and $\pi$ under flat priors (a derivation and further details of the EM algorithm are in S2 Note). Conditional on the MAP estimates $\widehat{\Sigma}_{1:S}, \widehat{\pi}$, the posterior for $\theta_{\cdot g}$ is available in closed form (see Methods–*MV model when data are missing*).

## Visual overview of results

For a global comparison between UV and MV models, we visualise the output of the UV and MV analyses via $z$-statistics defined as $z{:=}\widehat{\theta}/\widehat{s}$. For enhanced interpretation, $z$-statistics are scaled by their corresponding significance threshold, i.e., we plot $\widetilde{z}{:=}z/\tau$, so that $|\widetilde{z}| \geq 1$ corresponds to a significant perturbation (Fig 2). There is a greater proportion of significant $z$-statistics in the MV model, with significance often co-occurring across phenotypes in the same procedure, and a tendency for direction to be correlated within procedure. Instances of missing data denoted by white regions in the UV model heatmap.

We go on to present heatmaps of the estimates of correlation corresponding to $\widehat{\Sigma}$ and $\widehat{R}$ (Fig 3). There are obvious blocks of correlation within several procedures, which is expected as similar phenotypes tend to cluster in procedures. While almost all of the experimental correlation in $\widehat{R}$ occurs between phenotypes within the same procedure, there is a substantial off block-diagonal correlation structure in $\widehat{\Sigma}$, indicative of correlated phenotypic perturbations across different procedures. For example, in Fig 3A, KO–gene perturbations are correlated between Open Field and Light–Dark Test, Clinical Chemistry and Body Composition, Auditory Brain Stem Response and Acoustic Startle, Body Composition and Echo, and Hematology and Clinical Chemistry.

## Power to detect KO perturbations

We first compare the statistical power of the MV and UV models to detect perturbations, i.e., call hits, at gene–phenotype pairs where data are observed, and so at which both UV and MV results are available. We control the Fdr below 5% using the Westfall–Young permutation procedure [31,32] based on specially created synthetic null data; details are in Methods–*Control of error rates*.

Fig 4 visually represents the relative power and overlap of the various methods, including comparing to the existing IMPC database, which contains results from a different UV approach [12]—we will discuss this comparison in more detail in Results–*Comparison with IMPC database*. There are 4,256 (1.4% of 302,997 observed data measurements) hits called by the UV model, compared to 31,843 (10.5%) hits in the observed data results of the MV model, corresponding to an estimated 7.5-fold increase in power of the MV model relative to the UV model. When we examine concordance between the UV and MV results, there are 95 (0.0%) hits called by the UV model only, compared to 27,682 (9.1%) hits called by the MV model only. Of the 4,256 UV model phenotype hits, the MV model co-calls a hit with the same directionality in 4,161 (97.8%) cases and never calls a hit in an opposite direction to the UV model.

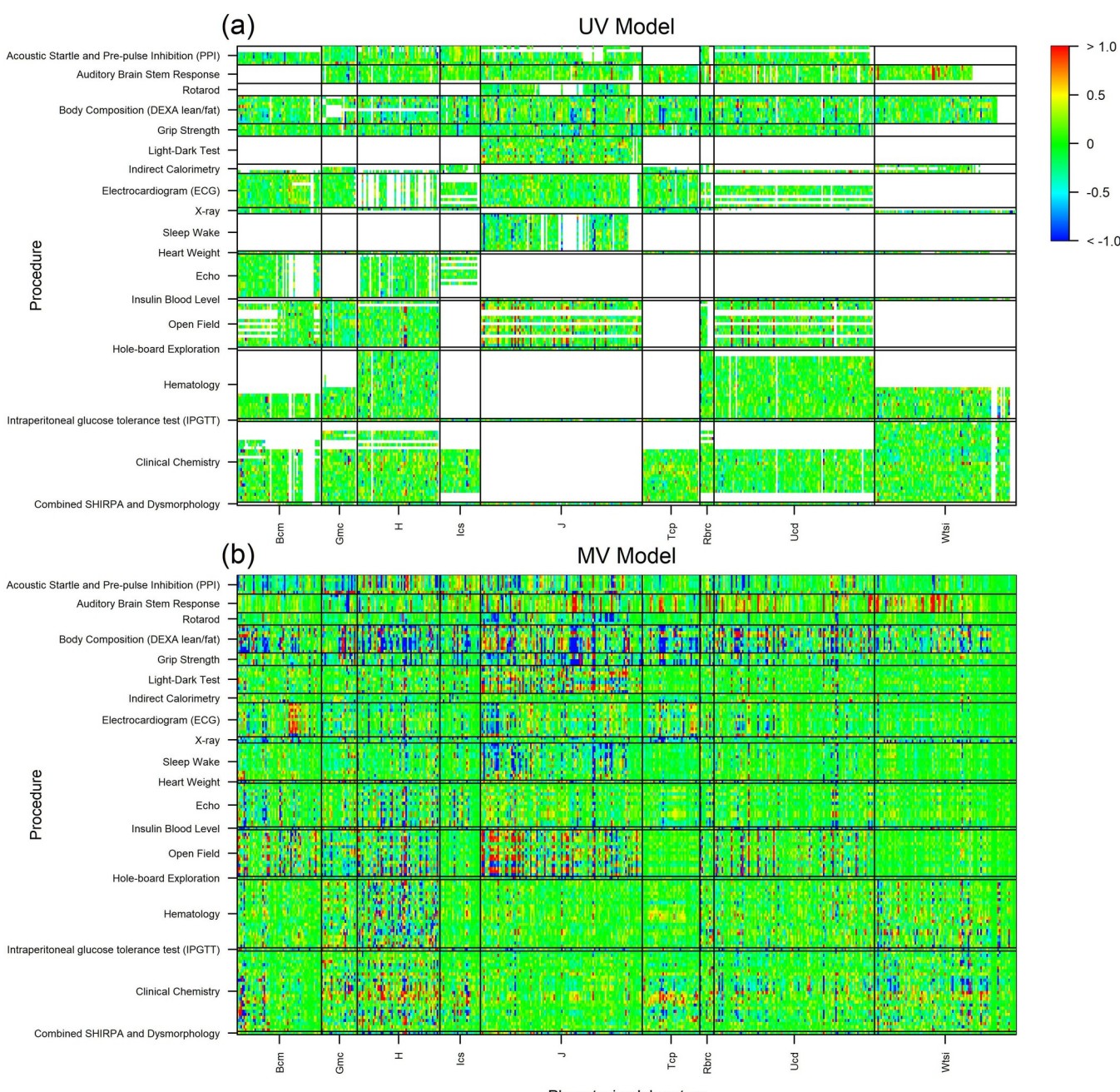

**Fig 2. Global representation of increased sensitivity of the MV model.** Each row corresponds to a phenotype, with multiple phenotypes grouped by procedure, labelled left. Each column corresponds to a KO line, with multiple lines grouped by phenotyping laboratory, labelled bottom. For effective visualisation, only a random subset of 500 KO lines is shown. The heatmaps display scaled $z$-statistics, so that $\tilde{z} > 1$ and $\tilde{z} < -1$ correspond to a gene KO causing a significant increase/decrease respectively in the phenotype (Methods–*Control of error rates*). (a) UV model, where white squares indicate missing phenotype data; and (b) MV model. The data and code used to generate this figure are available at [13,14]. KO, knockout; MV, multivariate; UV, univariate.

See also S3 Fig for a scatterplot comparing the scaled $z$-statistics outputted by the UV and MV models.

We go on to examine the relative sensitivity of the MV and UV models in more detail, by comparing the number of hits called by the 2 models at each phenotype (Fig 5A) and KO gene

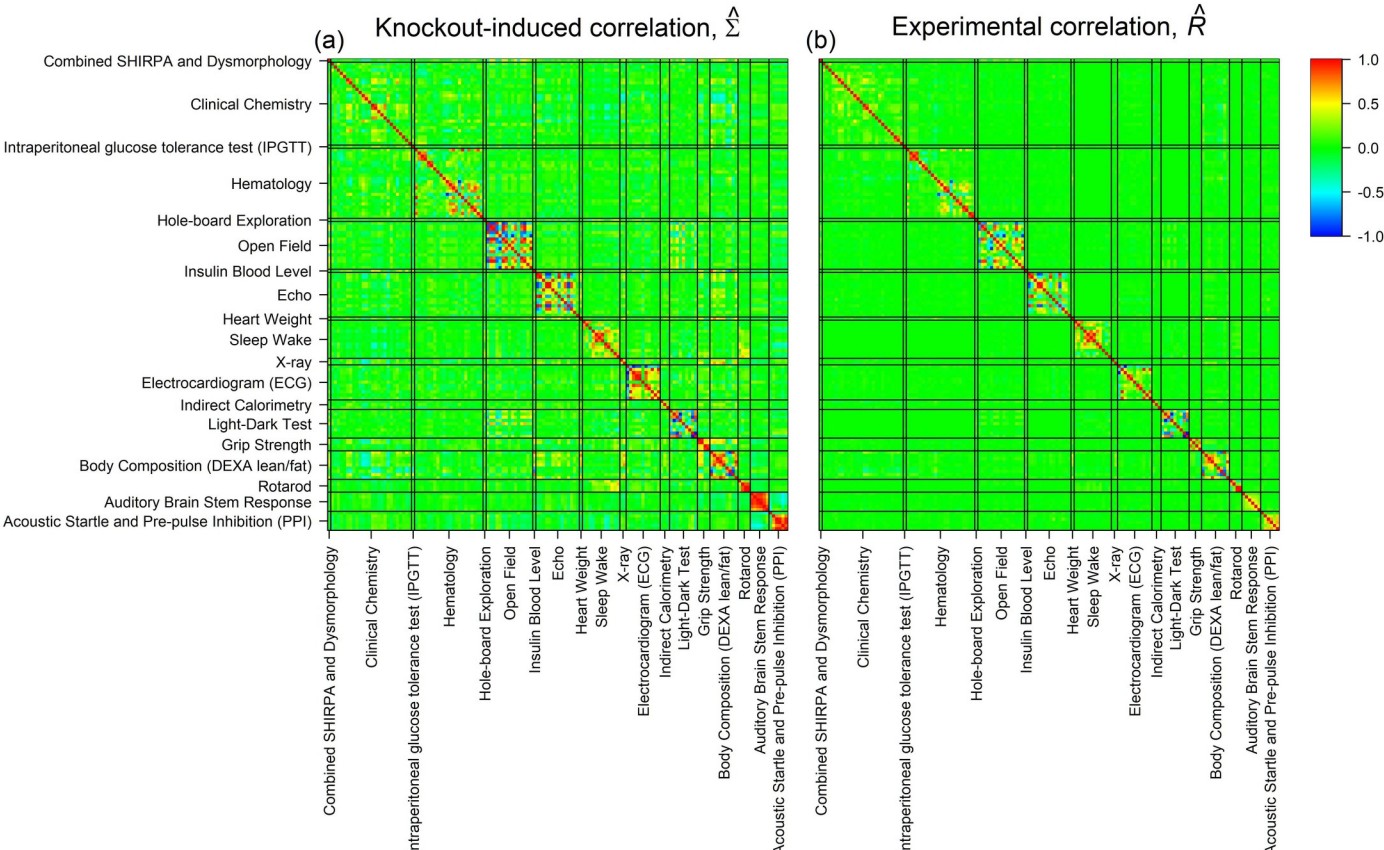

**Fig 3. Heatmaps of correlation matrices underlying a systematic co-perturbation of phenotypes in the IMPC.** (a) Estimated correlation matrix for the biological covariation induced by gene KOs, $\widehat{\boldsymbol{\Sigma}}$. (b) Estimated experimental correlation matrix, $\widehat{\boldsymbol{R}}$, attributable to the measurement process rather than the targeted biology. The data and code used to generate this figure are available at [13,14]. IMPC, International Mouse Phenotyping Consortium; KO, knockout.

(Fig 5B). The MV model identified more perturbations than the UV model in all 148 phenotypes and in 2,750 (60.5% of) KO lines; the UV model identifies more perturbations in 33 (0.7% of) KO lines. On average, the MV model calls 186.4 more hits per phenotype and 6.1 more hits per KO line than the UV model. Fig 5C examines the procedure-wise power enhancement of the MV method, presenting the proportion of KO lines that have at least one significantly perturbed phenotype in each procedure (see also S4 Fig for the phenotype-wise comparison, as well as details of the proportion of missing data for each phenotype).

## Inference when data are missing

Even for gene–phenotype pairs at which no data are measured, referred to here as *missing data*, the MV model can be used to infer gene KO effects via the correlation structure that exists between unmeasured and measured phenotypes. The MV model identifies perturbations in 4,819 (1.3% of 370,107 *missing data* cases), which compares favourably with the UV model's hit rate of 1.4% on *observed* data. When missing data results are combined with the observed data results, the MV model detects a total of 36,662 perturbations, an 8.6-fold increase compared to 4,256 detected by the UV method.

It is important to note that estimation of $\boldsymbol{\theta}_{\cdot g}$ when $\widehat{\boldsymbol{\theta}}_{\cdot g}^{\mathrm{UV}}$ is only partially observed can be performed coherently provided the statistical model is well specified with respect to the

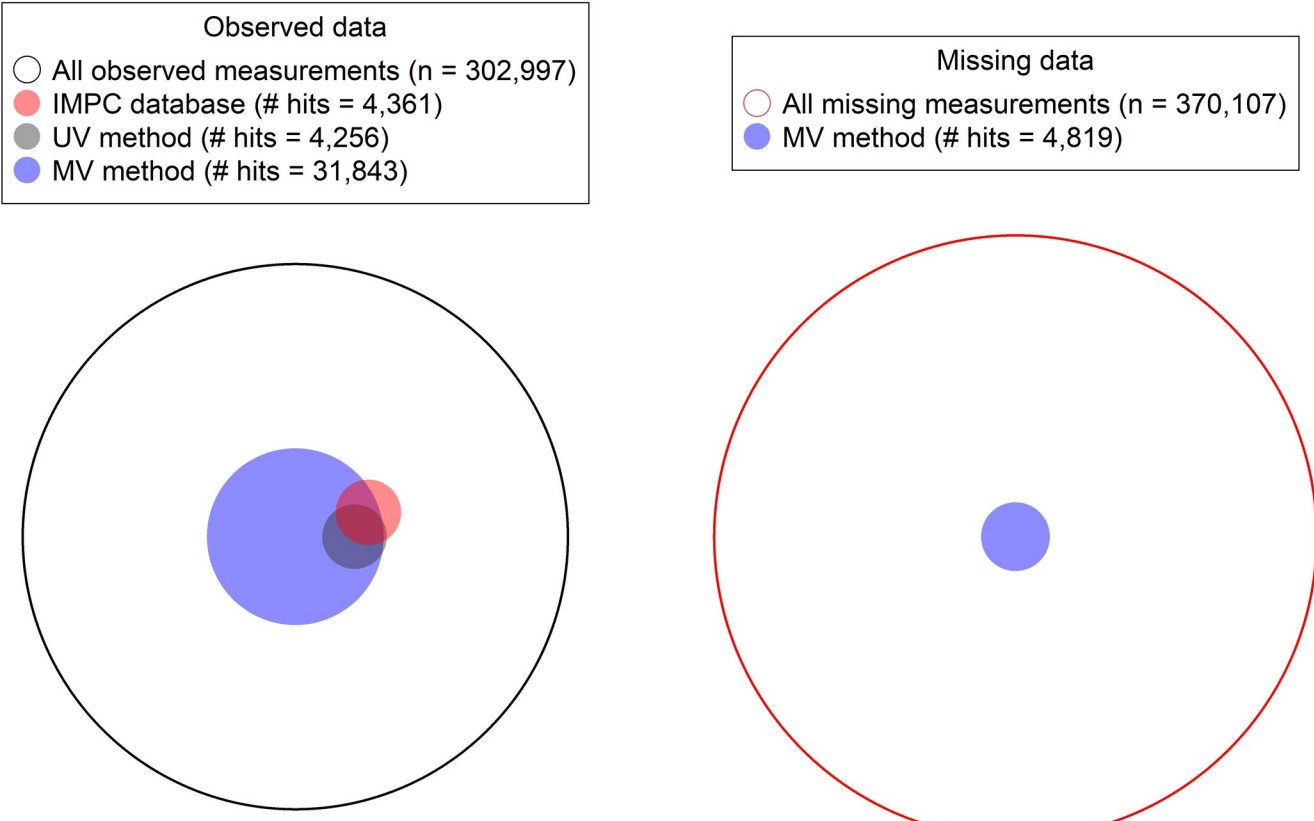

**Fig 4. Visual comparison of the methods' hit rates and overlap for observed and missing measurements.** The large black and red outlined circles denote, by area, the number of observed and missing measurements. Each of the circles corresponding to a method has area representing the number of hits called (on observed or missing data). The overlapping area between circles represents the number of hits called by both methods. The data and code used to generate this figure are available at [13,14]. IMPC, International Mouse Phenotyping Consortium; MV, multivariate; UV, univariate.

underlying data generating mechanism, and the unobserved data are missing at random (MAR) [23,33,34]. While there is a large proportion of missing data, it is clear from Fig 2 that the bulk of data is missing in obvious blocks and is a result of certain measurements/procedures not being performed in some centres. In this context of certain centres systematically not performing a subset of measurements, the MAR assumption is reasonable, in that the missing data mechanism, "given the missing data and the value of the observed data, is the same for all possible values of the missing data." [33].

In spite of this reassuring observation, there is naturally still going to be some relatively small proportion of data that violate the MAR assumption in such a large and complex data set as this. We therefore perform additional checks on how practically reasonable the MAR assumption is. These are described in Results–*Validating replicability* (with reference to Fig 6C and 6D), and in Methods–*Predicting masked data*. Our recommendation to practitioners is carefully to examine the appropriateness of the MAR assumption in their particular context in the light of the work of Rubin and colleagues [33,34]. If there are any doubts about the MAR assumption's validity, we recommend further empirical checks. In particular, the cross-validated mask and predict approach described in Methods–*Predicting masked data* can be implemented in a wide variety of MV datasets with missing data, and we recommend this as a tool for checking accuracy post hoc when the rate of missingness is high.

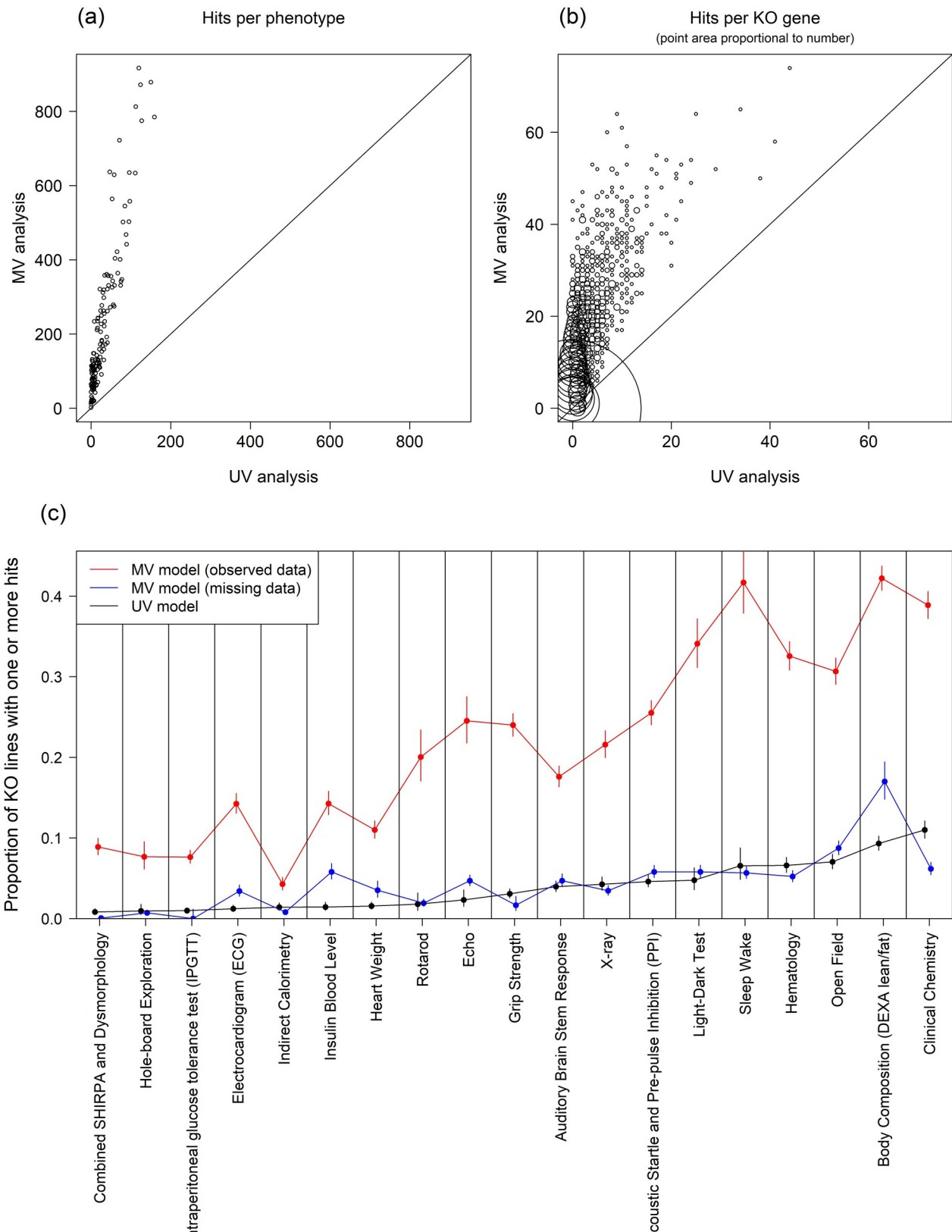

**Fig 5. Power enhancement: The MV method offers increased sensitivity to detect gene-KO-induced perturbations.** (a) Number of perturbations per phenotype identified by MV vs. UV models. (b) Number of perturbations per KO line identified by MV vs. UV models. (c) The proportion of lines with at least 1 hit in a procedure (i.e., having at least 1 phenotype perturbed in that procedure) is used to compare the power of the UV method

and MV method (on measured and missing data). Procedures are ordered by the UV model's hit rate. The data and code used to generate this figure are available at [13,14]. KO, knockout; MV, multivariate; UV, univariate.

## Validating replicability

We validate the UV and MV results by leveraging the multilaboratory nature of the experimental data. As part of the IMPC, a small number of KO lines have been measured multiple times across several labs, blind to their special status, i.e., the same gene KO, phenotyped in multiple labs; we refer to these as the *reference lines*. We analyse them under the UV and MV models while ensuring the models are blind to their correspondence to one another as replicated samples. After analysis, we reveal the reference lines and examine the replicability of findings on the same reference line across multiple phenotyping centres.

S5 Fig plots the annotation results for the reference lines under the UV and MV models. This visually reinforces the impact of the MV model: It strongly increases the hit rate (denoted by a higher density of crosses) and does so in a replicable way. The directionality of pair-wise reference line hits of the MV model is concordant in 295 cases and discordant in 7 cases. Observed levels of replicability can be usefully interpreted in terms of a corresponding false sign rate (Fsr), described in Methods–*Replicability and false sign rates* and estimated using the IMPC reference lines. We attain a reassuringly low global estimate of $\widehat{\text{Fsr}}_{\text{replicate}}$ = 1.2% (95% CI: 0.6% to 2.4%) for the MV model.

Fig 6 provides further insights into the degree of replicability across laboratories in the reference line replicates. The blue/red shaded regions in each panel contain instances where results respectively agree/disagree across laboratories. The MV model (Fig 6B, 6C and 6D) identifies more perturbations than the UV model (Fig 6A) and does so with a high level of replicability, as measured by the small number of points in the red shaded regions and quantified by Fsr estimates ($\widehat{\text{Fsr}}_{\text{replicate}}$) shown at top of each panel corresponding to the results shown in that panel. Importantly, the level of agreement across laboratories is good regardless of whether the data were missing (Fig 6C) or measured (Fig 6B) or were measured in one laboratory but not in the other (Fig 6D).

## Comparison with IMPC database

We compare the signed phenotype calls of our UV and MV models to the existing calls in the IMPC database, which are based on a different UV method [12]. The hit rate in the relevant subset of the IMPC database is 1.9%, while our UV model hit rate is 1.4% and our MV model hit rate on measured data is 10.5%. It is not straightforward to make direct comparisons with the IMPC database hit rate, owing to differences in error rate control (nominal $p < 10^{-4}$ in the IMPC database versus Fdr < 5% for our UV and MV models). However, when we inspect the concordance of our methods with the existing database, we see good agreement (Table 1), pointing to effective error rate control in all cases. Our UV model agrees with the IMPC database in all cases where both call a significant phenotype hit. Our MV model disagrees with the IMPC database in only 3 cases (0.1% of instances where they both call a hit). We examine these disagreements in more detail in Methods, where we conclude there to be little evidence among these 3 cases of either model outperforming the other.

## Heterozygotes versus homozygotes

For some genes in the IMPC, both the heterozygote and homozygote KO lines are measured. It is biologically reasonable that heterozygote and homozygote phenotypic perturbations,

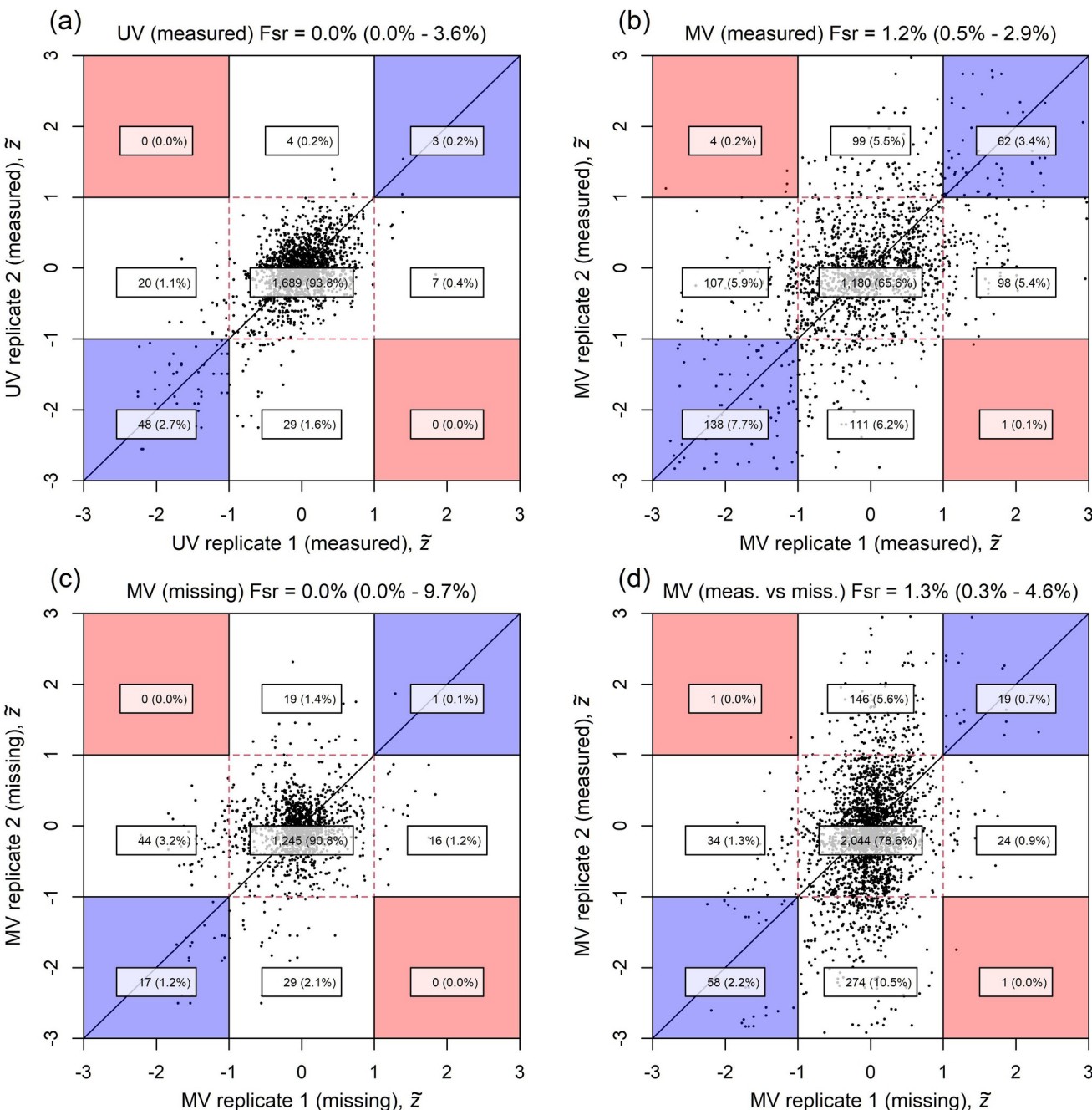

**Fig 6. Replicability validation scatterplots comparing results across phenotyping laboratories.** Each panel examines a different type of comparison of a pair of replication results: (a) UV model vs. UV model, (b) MV model (measured) vs. MV model (measured), (c) MV model (missing) vs. MV model (missing), (d) MV model (measured) vs. MV model (missing). We examine the interlaboratory agreement for the KO reference lines by scatterplotting scaled $z$-statistics, $\tilde{z}$, for the same KO line but measured in different laboratories. Significant perturbations correspond to $|\tilde{z}| > 1$, as delimited on the graphs with dashed red lines. Each point in the plots corresponds to 2 different laboratories measuring the same phenotype on the same KO line. The most informative cases for estimating the false sign rate (Fsr) occur when both laboratories detect significant perturbations, which correspond to points lying in the blue/red shaded regions on the scatterplot. The laboratories agree in the blue shaded regions but disagree in the red shaded regions. $\widehat{\mathrm{Fsr}}_{\mathrm{replicate}}$ estimates (95% CIs) are shown at the top of each panel and are based on the level of agreement/disagreement observed in the shaded regions (Methods–*Replicability and false sign rates*). Counts (%) for each significance combination are superimposed; while the axes extend to $[-3, 3]$, the counts apply to all data, including those beyond the plot's scale. The data and code used to generate this figure are available at [13,14]. Fsr, false sign rate; KO, knockout; MV, multivariate; UV, univariate.

**Table 1. Comparison of signed hits with the existing IMPC database.** (a) UV model; and (b) MV model. Each model is compared to the corresponding hits called in the existing IMPC database (top). We represent calls by a number in {−1,0,1}, with 1 and −1 denoting significant positive and negative phenotypic perturbations, respectively, and 0 denoting a lack of statistical significance.

| | | (a) UV Model | | | (b) MV Model | | |
|---|---|---|---|---|---|---|---|
| | | −1 | 0 | 1 | −1 | 0 | 1 |
| | −1 | 902 | 384 | 0 | 1,678 | 10,008 | 3 |
| IMPC database | 0 | 1,215 | 188,395 | 875 | 943 | 207,387 | 742 |
| | 1 | 0 | 364 | 458 | 0 | 7,075 | 995 |

should they exist, are likely to act in the same direction. We can therefore compare the heterozygote/homozygote pairs for concordance in results (S6 Fig). As is expected biologically, homozygote lines are called as hits more frequently by the MV model (7.6%) than the corresponding heterozygote lines (2.3%). In cases where both the heterozygote and homozygote lines for a gene are called as hits, we observe directional concordance in 594 cases and discordance in only 46 cases. Under the assumption that all heterozygote/homozygote pairs truly perturb the phenotype in the same direction, then this level of discordance is consistent with $\widehat{\mathrm{Fsr}}_{\mathrm{replicate}}$ of 3.7% (95% CI: 2.8% to 5.0%). This low Fsr estimate contributes further evidence that our control of false positive rates in hit calling is effective, adding to the evidence provided by the reference lines replicability analysis. In reality, there may be exceptions whereby the heterozygotes and homozygotes actually perturb the phenotype in different directions, in which case this zygosity-based estimate $\widehat{\mathrm{Fsr}}_{\mathrm{replicate}}$ may still be usefully interpreted as an upper bound on the actual $\mathrm{Fsr}_{\mathrm{replicate}}$.

## Gene ontology co-enrichment

Gene Ontology (GO) uses a directed graph to annotate and interrelate biologically meaningful GO terms [35,36] such as *sensory perception of sound* (GO:0007605) and *locomotory behaviour* (GO:0007626). Each GO term has its own gene set, a list of mouse genes assigned either by manual curation of published experimental literature or via automated computational methods. Analogously, our analysis of the IMPC database generates, for each IMPC phenotype, a set of genes that cause significant phenotypic perturbations; we say each IMPC phenotype has its own gene set.

By identifying GO terms and IMPC phenotypes with overlapping gene sets, we aim to increase understanding about the general biological characteristics of genes that affect a phenotype. Furthermore, observing co-enrichment between GO terms and IMPC phenotypes adds evidence that the statistical methods are performing well (assuming the false positive rate for detecting co-enrichment is controlled appropriately). In this section, we therefore ask the question: Which pairs of GO term gene sets and IMPC gene sets share a larger set of genes than expected by chance? An example of this type of co-enrichment analysis is presented in Fig 7, where we quantify the overlap between gene sets for GO:*locomotory behavior* and IMPC: *Locomotor activity*.

We test for co-enrichment between each of the 148 IMPC gene sets and 5,368 GO terms in the Biological Process (BP) Sub-Ontology that are annotated to one or more IMPC KO genes. We focus on genes exhibiting large perturbations ($\geq 2$ population SDs) and control the family-wise error rate at each phenotype below 5% for testing across all BP GO terms; see Methods–*Gene ontology analysis* for further details. Across all gene sets, the MV model identifies co-enrichment between 1,359 pairs of IMPC and GO gene sets, compared to the UV model, which shows co-enrichment at 342 pairs. The MV model identifies more co-enriched GO gene sets than the UV

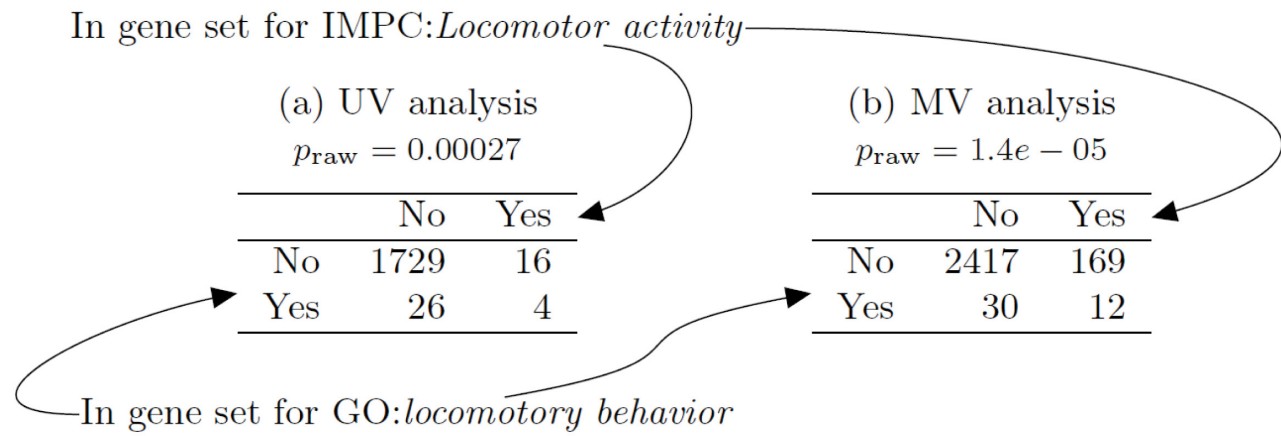

**Fig 7. Illustrative 2-by-2 contingency tables for co-enrichment testing.** (a) UV model; (b) MV model. Each contingency table allocates each gene to one of 4 categories according to whether it is in the GO term gene set (left) and/or IMPC phenotype gene set (top). Fisher exact test *p*-values are shown above each table. GO, Gene Ontology; IMPC, International Mouse Phenotyping Consortium; MV, multivariate; UV, univariate.

model at 80 (54% of) IMPC gene sets, while the UV model identifies more co-enriched GO gene sets at only 17 (11% of) IMPC gene sets (Table 2 presents a more detailed comparison).

Fig 8 provides a global characterisation of the systematic relationships between IMPC phenotypes and biological pathways. A comparison with the corresponding UV model's map (S7 Fig) illustrates the greatly increased number of GO annotations arising from the MV model, while also demonstrating qualitative agreement between the UV and MV results in cases where both models show gene enrichment. In Fig 8, the rows and columns are clustered on the basis of co-enrichment patterns. This clustering is performed without reference to the grouping of phenotypes by procedure, so it is remarkable that phenotypes from the same procedure tend to be clustered together in the horizontal direction (phenotype labels are coloured according to procedure—see legend at bottom left of Fig 8).

The global picture in Fig 8 is one of concordance between the co-enrichment analysis and existing scientific knowledge. To illustrate this, we now examine in detail a few representative rows of the heatmap in Fig 8, labelled (a-h), presented in more detail in the subtables of Fig 9.

- GO:*regulation of lipid biosynthetic process* is co-enriched with IMPC:*Total cholesterol*, IMPC:*HDL-cholesterol* and IMPC:*Triglyceride* phenotypes from the IMPC:*Clinical Chemistry* procedure (Fig 9A).

**Table 2. Co-enrichment counts compared across the UV and MV models.** (a) Number of GO terms co-enriched with each IMPC phenotype; e.g., there are 23 phenotypes that have 0 GO terms enriched for the UV model but which have between 1 and 5 GO terms enriched for the MV model. (b) Number of IMPC phenotypes co-enriched with each GO term; e.g., there are 180 GO terms that have 0 phenotypes enriched for the UV model but which have between 1 and 5 phenotypes enriched for the MV model.

| | | (a) GO terms per phenotype | | | | | (b) Phenotypes per GO term | | | | |
| --- | --- | --- | --- | --- | --- | --- | --- | --- | --- | --- | --- |
| | | MV model | | | | | MV model | | | | |
| | | **0** | **1–5** | **6–10** | **11–20** | **>20** | **0** | **1–5** | **6–10** | **11–20** | **>20** |
| | 0 | 50 | 23 | 6 | 15 | 7 | 5,006 | 180 | 15 | 7 | 0 |
| | 1–5 | 7 | 3 | 1 | 5 | 10 | 64 | 48 | 13 | 10 | 10 |
| UV model | 6–10 | 2 | 1 | 1 | 0 | 2 | 0 | 0 | 7 | 7 | 1 |
| | 11–20 | 1 | 0 | 5 | 4 | 4 | 0 | 0 | 0 | 0 | 0 |
| | >20 | 0 | 0 | 0 | 0 | 1 | 0 | 0 | 0 | 0 | 0 |

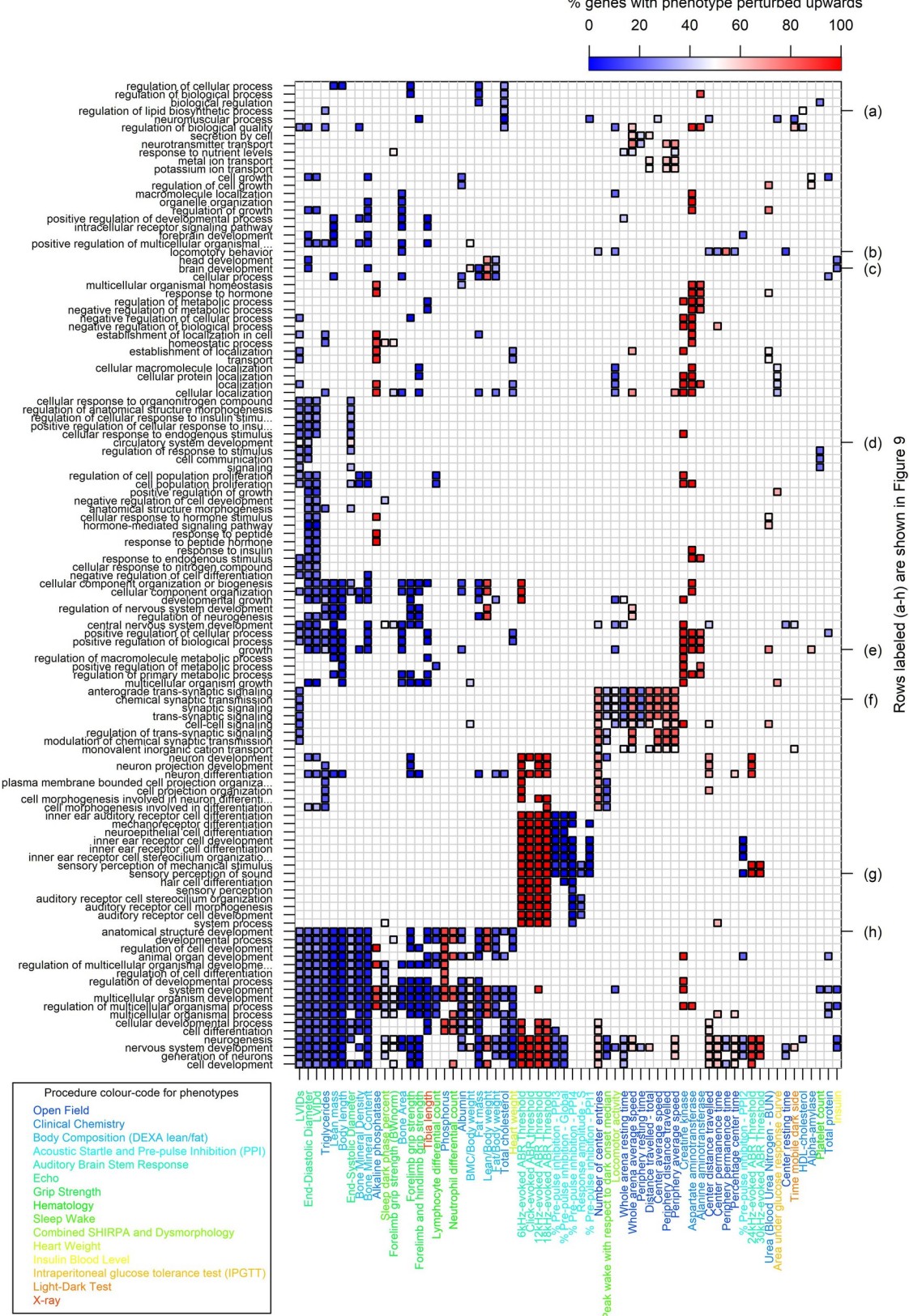

**Fig 8. Co-enrichment of GO terms (left) with IMPC phenotypes (bottom) for hits called by MV model.** Statistically significant co-enrichment between GO terms and IMPC phenotypes is denoted by bold outlined squares (controlling family-wise error rate < 5% for each phenotype). The colour of the square indicates the percentage of significantly perturbing KO genes at the GO term that change the phenotype in the positive direction (see scale bar at top). IMPC phenotypes are clustered by GO term pattern

along the horizontal axis, while BP GO terms are clustered vertically by phenotype pattern. Phenotype labels are coloured according to procedure as per legend at bottom left. A subset of GO terms, labelled by row (a-h) at right, are examined in more detail in Fig 9. For legibility, we only include in the plot those IMPC phenotypes and GO terms that have at least 3 instances of significant co-enrichment. The data and code used to generate this figure are available at [13,14]. BP, Biological Process; GO, Gene Ontology; IMPC, International Mouse Phenotyping Consortium; KO, knockout; MV, multivariate.

- GO:*locomotory behavior* is co-enriched with phenotypes from the IMPC:*Open Field* procedure, which is used to assess anxiety and exploratory behaviours; IMPC:*Bone Area* also shows overlap with GO:*locomotory behavior*, compatible with abnormal bone structure contributing to impaired movement (Fig 9B).

- GO:*brain development* displays interesting overlap with metabolic phenotypes from the IMPC:*Body Composition (DEXA lean/fat) (DEXA)* procedure and also with IMPC:*Insulin* (Fig 9C).

- GO:*circulatory system development* overlaps with heart-function phenotypes from the IMPC:*Electrocardiogram (ECG)* procedure (Fig 9D).

- GO:*growth* and GO:*anatomical structure development* are co-enriched with a broad range of IMPC phenotypes representative of systemic perturbation affecting body size, strength and metabolism (Fig 9E and 9H.

- GO:*chemical synaptic transmission* is co-enriched with phenotypes from the IMPC:*Open Field* procedure, thereby pointing to the connection between synaptic dysfunction and impaired movement, anxiety and exploratory behavioural phenotypes (Fig 9F).

- GO:*sensory perception of sound* is co-enriched with IMPC phenotypes are mainly in the IMPC:*Auditory Brain Stem Response (ABR)* and IMPC:*Acoustic Startle and Pre-pulse Inhibition (PPI)* procedures. This makes sense as the ABR procedure [37] directly targets hearing sensitivity, while the PPI procedure is largely used to assess sensorimotor gating (the ability of a sensory event to suppress a motor response) [38] (Fig 9G).

To close the GO co-enrichment section, we note that this type of analysis will have greatest power and provide optimal insight once all gene KOs in the IMPC have been phenotyped and the data analysed. Our discussion here is intended just to give a flavour of what insights will be provided by the final analyses of the complete data set.

## Factor analysis of MV perturbations

An eigendecomposition of the MV model's fitted covariance structure ($\widehat{\boldsymbol{\Sigma}}_{\text{pooled}}$ of (26) in Methods–*Cross-validation and model averaging*) shows that 75% of the correlation structure is explained by the first 20 eigenvectors; S8 Fig plots the cumulative variance explained. We rotate these eigenvectors to a sparse, interpretable set of loadings, or *factors*, which are visualised in Fig 10A. The important notion of sparsity in this context, illustrated in Fig 10A, is that the vast majority of phenotypes at any particular factor have loadings close to zero (i.e., they are coloured green). Each factor therefore defines a small cluster of phenotypes that have large positive or small negative loadings. From a biological perspective, each cluster of phenotypes tends to be perturbed collectively by gene KOs.

By examining each cluster of phenotypes, and taking into account the signs of its loadings, we manually curate labels describing the biological interpretation of each factor. For example, the first factor defines a cluster according to negative loadings on Bone Mineral Content, Bone Area, Lean mass, Body length, and Heart weight; this factor is therefore labelled "Body size

(a) GO gene set *regulation of lipid biosynthetic process*

| IMPC gene set | Co-enrich p |
| --- | --- |
| HDL-cholesterol | 1.2e-06 |
| Total cholesterol | 2.2e-06 |
| Triglycerides | 2e-05 |

(b) GO gene set *locomotory behavior*

| IMPC gene set | Co-enrich p |
| --- | --- |
| Percentage center time | 3.2e-06 |
| Center distance travelled | 4.8e-06 |
| Locomotor activity | 1.7e-05 |
| Number of center entries | 2.5e-05 |
| Center permanence time | 2.8e-05 |
| Periphery permanence time | 2.9e-05 |
| Bone Area | 3.4e-05 |
| Center resting time | 5.3e-05 |

(c) GO gene set *brain development*

| IMPC gene set | Co-enrich p |
| --- | --- |
| Lean/Body weight | 2.8e-06 |
| End-Diastolic Diameter | 2.9e-06 |
| Fat/Body weight | 8e-06 |
| Insulin | 1.9e-05 |
| Fat mass | 2.7e-05 |
| BMC/Body weight | 3.3e-05 |
| Bone Mineral Content | 5e-05 |

(d) GO gene set *circulatory system development*

| IMPC gene set | Co-enrich p |
| --- | --- |
| End-Systolic Diameter | 3.4e-06 |
| LVIDs | 5.5e-06 |
| End-Diastolic Diameter | 6e-05 |

(e) GO gene set *growth*

| IMPC gene set | Co-enrich p |
| --- | --- |
| Bone Mineral Content | 5.6e-09 |
| Bone Area | 4.1e-07 |
| End-Diastolic Diameter | 7e-07 |
| LVIDd | 2.5e-06 |
| Aspartate aminotransferase | 3.9e-06 |
| Lean mass | 5.1e-06 |
| Body length | 5.4e-06 |
| Creatine kinase | 5.8e-06 |
| Albumin | 1.7e-05 |
| Triglycerides | 1.9e-05 |
| Urea (Blood Urea Nitrogen - BUN) | 2e-05 |
| Alpha-amylase | 2.9e-05 |
| Bone Mineral Density | 3e-05 |
| Alanine aminotransferase | 4.4e-05 |
| Locomotor activity | 5.4e-05 |

(f) GO gene set *chemical synaptic transmission*

| IMPC gene set | Co-enrich p |
| --- | --- |
| Peak wake with respect to dark onset median | 8.8e-07 |
| Periphery average speed | 2.5e-06 |
| Center average speed | 2.7e-06 |
| Number of center entries | 3e-06 |
| Whole arena average speed | 5.6e-06 |
| Whole arena resting time | 7.7e-06 |
| Locomotor activity | 8.3e-06 |
| Distance travelled - total | 9.1e-06 |
| Periphery resting time | 1.3e-05 |
| Periphery distance travelled | 1.8e-05 |
| LVIDs | 2.6e-05 |

(g) GO gene set *sensory perception of sound*

| IMPC gene set | Co-enrich p |
| --- | --- |
| 12kHz-evoked ABR Threshold | 4.9e-11 |
| 18kHz-evoked ABR Threshold | 5.6e-11 |
| Click-evoked ABR threshold | 1.2e-10 |
| 6kHz-evoked ABR Threshold | 2.1e-10 |
| % Pre-pulse inhibition - PPI4 | 2.6e-08 |
| 24kHz-evoked ABR Threshold | 3.2e-08 |
| 30kHz-evoked ABR Threshold | 1.7e-06 |
| % Pre-pulse inhibition - PPI1 | 3.6e-06 |
| Response amplitude - S | 8.9e-06 |
| % Pre-pulse inhibition - Global | 1.5e-05 |
| % Pre-pulse inhibition - PPI3 | 2e-05 |
| % Pre-pulse inhibition - PPI2 | 3.3e-05 |

(h) GO gene set *anatomical structure development*

| IMPC gene set | Co-enrich p |
| --- | --- |
| Lean mass | 8.9e-09 |
| Bone Mineral Content | 2.4e-08 |
| Tibia length | 4.6e-08 |
| Triglycerides | 5.2e-08 |
| Neutrophil differential count | 1.1e-07 |
| End-Diastolic Diameter | 1.3e-07 |
| Bone Mineral Density | 1.4e-07 |
| LVIDd | 2.4e-07 |
| Body length | 2.6e-07 |
| Fat mass | 3.7e-07 |
| Fat/Body weight | 5.7e-07 |
| Phosphorus | 2.2e-06 |
| Lean/Body weight | 2.8e-06 |
| Albumin | 3.1e-06 |
| Total cholesterol | 3.3e-06 |
| End-Systolic Diameter | 3.6e-06 |
| LVIDs | 4.7e-06 |
| Heart weight | 4.7e-06 |
| Forelimb grip strength | 8.3e-06 |
| Lymphocyte differential count | 1.6e-05 |

**Fig 9. Examples of GO and IMPC gene set co-enrichment.** Each table lists the instances of significant co-enrichment between a GO term (labelled top) and IMPC phenotypes (left column), along with Fisher exact test *p*-values quantifying evidence for co-enrichment (right column). GO, Gene Ontology; IMPC, International Mouse Phenotyping Consortium.

(−).” Factor labels are shown on the axes of Fig 10B. The suffix (+) or (−) denotes the direction-ality of effect implied by the sign of the loadings in Fig 10A, e.g., "Body size (−)" denotes "Reduced Body size" and "Deafness (+)" denotes "Increased deafness." We test for perturba-tions in the scores corresponding to each factor (Methods–*Factor model*); Fig 10C plots the proportion of KO lines with significantly perturbed factor scores, separated according to whether the perturbation is positive or negative (along the axis defined by the corresponding loadings vector in Fig 10A). The factors are ordered according to the proportion of lines with significant perturbations; the commonest perturbation is identified in 27.5% of lines while the least common is identified in 3.7%. The signs of the loadings vectors are defined so that the majority of perturbations are positive, resulting in an average of 69.1% in the positive direction.

We characterise the statistical co-perturbation of factors by analysing a 2×2 contingency table of perturbation significance counts for each pair of factors, where here factor perturba-tions are stratified in the binary form {0 ≡ factor is not significantly perturbed, 1 ≡ factor is significantly perturbed}. There is significant evidence of co-perturbation in almost every case; specifically, the null hypothesis of independence of perturbation across pairs of factors is rejected in 187 out of 190 cases (Fisher exact test on 2×2 tables with Fdr controlled at 5%). Fig 10B displays odds ratios (ORs) quantifying the statistical co-perturbation of each pair of fac-tors. Here, the interpretation of the OR between a particular pair of factors is that observing a perturbation in one factor multiplies the odds of observing a perturbation in the other factor by OR. In Fig 10B, there are some groups of factors that tend to be relatively strongly co-per-turbed, for example, (Body size, Cardiac dysfunction); (Activity/exploration factors, Coordina-tion/balance, Sleep bout length, Sleep daily percent). Some factors, such as Activity/exploration 3, Sleep daily percent, Neutrophil:lymphocyte ratio are strongly co-perturbed with a number of other factors. Others, such as Deafness are less strongly co-perturbed with other factors (indicated by ORs relatively close to 1 in the Deafness (+) row of Fig 10B).

## Discussion

The IMPC has revealed a clear dependence structure in KO-induced phenotypic perturba-tions. Here, we have demonstrated that some of the correlation is attributable to multiple mea-surements of a single underlying phenotype (mainly within-procedure correlation in Fig 3A) and some of it is attributable to pleiotropic gene effects (particularly some of the between-pro-cedure correlation in Fig 3A). Given this structure, it is to be expected that sharing information across phenotypes can greatly aid annotation. Performing MV analysis in this context is chal-lenging, not least because of the size of the data set and the complex intersample correlation structures induced by the experimental design. We have developed a composable 2-stage MV modelling approach that addresses these issues.

The increase in hit rate from 1.4% for the UV model to 10.5% for the MV in the measured data setting is noteworthy. The MV model's hit rate of 1.3% in the case of missing measure-ment data is practically useful when compared to the UV model's 1.4% on observed data. To verify the validity and coherence of our results, we implemented several separate measures and checks. An essential element of our approach is the generation of realistic synthetic null lines through in silico relabelling of contemporaneously measured WT animals. These synthetic nulls underpin several analyses, including enhancing the estimation of the experimental corre-lation *R*, but their most vital role is the calibration of false positive rates in hypothesis testing for phenotype annotation. This particular application underscores the importance of blinded phenotyping of control animals in phenotyping pipelines, and the utility of WT animals

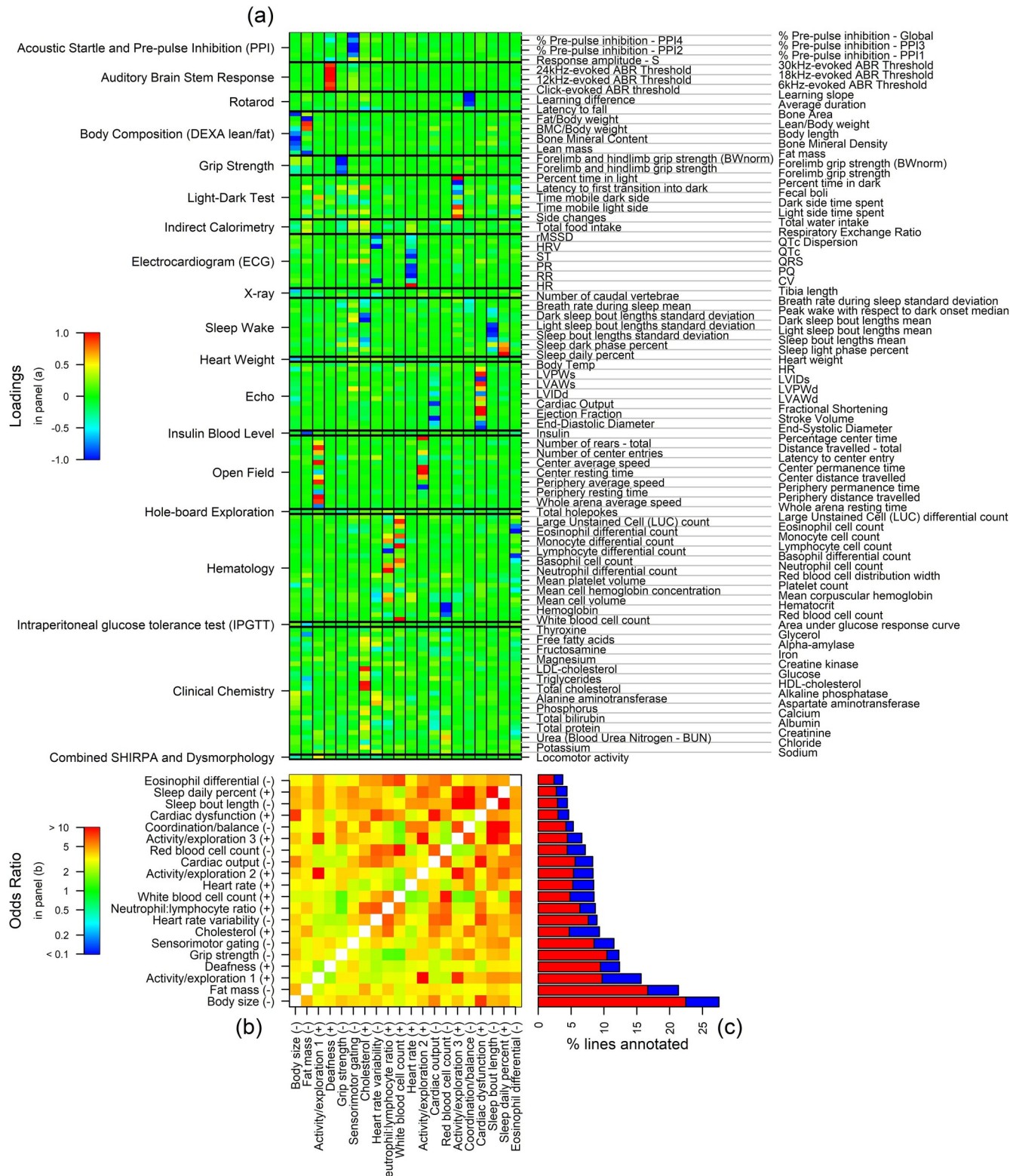

**Fig 10. Characterisation of sparse factors underlying $\widehat{\Sigma}$. (a)** Sparse loadings for 20 factors; each loadings vector is signed and scaled so that the magnitude of the largest loading is 1 and >50% of significant factor score perturbations are positive, as can be seen in panel (c). **(b)** Odds ratio as a measure of dependence in perturbations between pairs of factors (Results–*Factor analysis of MV perturbations*). **(c)** The hit percentages (i.e., percentage significantly perturbed) at each factor, with red/blue, respectively, indicating the percentage of lines perturbed in the same/opposite direction to the loadings. For example,

the large proportion of red in the bar labelled "Body size (−)" indicates that most perturbations are in the same direction as the factor loadings, i.e., they tend to result in *reduced* Body size. The data and code used to generate this figure are available at [13,14].

following the same experimental design as KO animals, for example, in the sharing of litters, days, and other experimental covariates.

There have been historical concerns over the replicability of animal phenotyping annotations across different laboratories, particularly with behavioural phenotypes [39,40]. Here, we have focused on replicability through the lens of signed annotations in {−1,0,1}, with ±1 corresponding to statistically significant perturbations in a particular direction, and 0 representing no significant effect. Signed annotations are vital for the effective scientific impact of the IMPC, and their replicability is therefore a fundamental downstream requirement of any statistical method. In the context of phenotypic screening in model species, sample size, and hence power, is strictly limited. We therefore expect in our replicability study to observe many reference line phenotype hits called in one laboratory but not in another. However, when 2 laboratories both call hits, the proportion of hit pairs that are concordant is a useful measure of replicability. Here, we have shown that the level of concordance, and hence replicability, in the IMPC reference lines is high, in particular showing compatibility with a low Fsr of $\widehat{\mathrm{Fsr}}_{\mathrm{replicate}} =$ 1.2% (95% CI: 0.6% to 2.4%). Notably, pair-wise concordant hits across laboratories were observed in all procedures in Fig 6 and S5 Fig, including behavioural ones such as Open Field and Light–Dark test. This check on coherence was only feasible because of the inclusion of reference lines and demonstrates the value of experimental design that incorporates technical replication across potentially heterogeneous measurement contexts.

As a further coherence check, we quantified directional concordance in phenotype hits between heterozygotes and homozygotes of the same KO line. Here, the agreement observed in the results of the MV model was compatible with an Fsr estimate $\widehat{\mathrm{Fsr}}_{\mathrm{replicate}}$ of 3.7% (95% CI: 2.8% to 5.0%), even though the observed level of discordance here is potentially inflated by any heterozygote/homozygote KOs having truly opposing effects.

We have demonstrated under the MV model that it is possible to call hits with relatively high power for gene–phenotype pairs at which measurements were not taken. This has the potential to enhance the scientific impact of the IMPC database, as a complete gene-by-phenotype annotation matrix offers a more encyclopaedic and versatile tool to end users, compared to a matrix with blocks of missing data (Fig 2). We have assessed the accuracy of inference in the presence of missing data in a number of ways. For the reference lines, the replicability of results on missing data is comparable to that of results on measured data (Fig 6). Separately, we have demonstrated high power (Fig 5 and S4 Fig) and accuracy (S9 Fig) of inference for whole missing procedures via a leave-one-procedure-out (LOO-MV) cross-validation technique. These results suggest a degree of redundancy in the data, including across procedures, in the sense that most of the information in some phenotypes is captured in others. The missing data methods developed here have the potential to replace some animal experiments with statistical analysis, in line with the NC3Rs [24]. It would be particularly effective if costlier experiments could be rendered redundant by less costly ones, where cost encompasses the ethical cost of animal suffering as well as considerations of finance and other resources.

We have focused on 148 quantitative phenotypes, yet the IMPC additionally includes many categorical phenotypes. Usefully, our MV model is straightforward to extend to multiple phenotypes of mixed response type. This is because it accepts as input the UV-estimated effect sizes and standard errors, and these can just as easily take the form of estimated log ORs outputted by a logistic regression as they can estimates and standard errors from an ordinary

linear model. We anticipate extending and applying the methodology in this direction, beginning with UV analyses using generalised linear multilevel models.

An MV model on estimated effect sizes and standard errors, as introduced in [22,23], has a number of benefits that may render it useful in other areas of application. The initial UV analyses may be made arbitrarily complex, allowing careful UV modelling of correlation structure across samples; there is no need at the UV stage to simultaneously consider the correlation structure across different response variables, which would be difficult both analytically and computationally; UV model fits may be performed in parallel; and the size of the data set inputted into MV analysis is reduced substantially, potentially by an order of magnitude or more. There are certain dataset properties that are preferred for the fruitful application of this method. We require a sufficient number of independent MV observations (in our case KO lines) to estimate the covariance structure in $\Sigma$ and $R$ effectively. If there are insufficient data to estimate full $P \times P$ covariance matrices, then $\Sigma$ and $R$ can be represented more parsimoniously using reduced rank factor models, as we do here. Misspecification of either the UV model correlation structure (in our case across animals) or of the independence assumption in the MV model (in our case of i.i.d. effect vectors across KO lines) may lead to miscalibrated output. We calibrated our model output using permutation-generated synthetic null lines, and we expect this would be a useful if not essential element in other applications as well.

The empirical Bayes approach to inference has major advantages, in that it allows the computer-intensive work, of estimating $\Sigma$ and $R$ and defining appropriate significance thresholds, to be done in advance. Thereafter, it is computationally tractable and fast to update estimates $\widehat{\theta}_{pg}^{\mathrm{MV}}$ as data on new KO lines, or further measurements on existing lines, become available. An alternative approach would be to perform full Bayesian inference targeting the posterior $p(\Sigma, R, \Theta | \widehat{\Theta}^{\mathrm{UV}})$. We did implement full Bayes by MCMC but found it to be less practicable than MAP estimation followed by empirical Bayes inference; this was mainly due to concerns about slow mixing of the MCMC sampler.

An even more ambitious goal would be to fit a full multilevel factor model directly to the raw data [41,42,43,44,45], i.e., to target the posterior $p(\Sigma, R, \Theta | Y)$, where $Y$ is the raw, animal-level data. This is in principle extremely attractive, as it would potentially allow for more information to flow the raw data to the parameters of interest and could deliver more power. Of course, effective inference would rely upon the (more complex) model being a sufficiently good representation of the data. With reference to the multilevel UV model at (1), a joint multilevel factor model would probably require an intermeasurement ($P \times P$) covariance structure underlying each of the different random effects; this would be nontrivial to implement, especially with nonidentifiability considerations. A more basic challenge is the size of the data set increasing by an order of magnitude, which could have a considerable impact on the computational complexity, depending upon the implementation. While the scope of this paper is to build on and extend the modular framework of [22,23], we do see joint multilevel factor modelling as a promising area to explore in future, especially with the ongoing development of scalable optimisation methods for complex models [46].

The development of a sparse factor model reduces the dimensionality of the space containing 75% of MV perturbations from 148 to 20. Even within that reduced 20-dimensional space, we observed strong interfactor correlations in annotation, suggestive of a still smaller effective dimensionality. In this latent space of factors, it is easier to place particular KO lines into a broader context. For example, we can identify which factors are perturbed and examine their particular properties, such as their biological interpretation, how rare the perturbation is in the IMPC more generally, and whether a perturbation's directionality is common or rare (positive or negative effect in Fig 10C).

In summary, we have developed a composable MV approach for analysis of high-dimensional data sets from the IMPC, demonstrating 4 major improvements over existing UV methods. First, power to detect KO perturbations can be increased drastically by purely analytical means, yielding 7.5 times as many gene–phenotype hits on observed data. Second, even when KO lines are missing some measurements, we can call hits at missing measurements with good power and output a full gene–phenotype map. Third, the greater power of the MV approach enhances correspondence between IMPC phenotypes and existing GO databases, promising ever stronger biological insights as the IMPC database progresses towards completion. Finally, high-dimensional phenotype perturbations may be informatively viewed in a much smaller, here 20-dimensional, subspace, thereby facilitating interpretation of gene KO effects and illuminating a rich structure in the phenotypic landscape of the mouse genome.

## Methods

### MV model when data are missing

Here, we generalise the MV model introduced at (2)–(4) to the case in which some subset of measurements is not observed. The validity of the MV missing data model below relies on the data set satisfying the MAR assumption [23,33,34] discussed in Results–*Inference when data are missing*. If, at gene $g$, only $P_g \leq P$ measurements are observed then, using the $*$ subscript $\widehat{\boldsymbol{\theta}}_{*g}^{\mathrm{UV}}$, $\boldsymbol{\theta}_{*g}$, $\boldsymbol{R}_{**}$ to denote restriction to the $P_g$ indices of the measured data, the model with missing data is written

$$p(\widehat{\boldsymbol{\theta}}_{*g}^{\mathrm{UV}}|\boldsymbol{R}) = \mathrm{N}(\widehat{\boldsymbol{\theta}}_{*g}^{\mathrm{UV}}|\boldsymbol{\theta}_{*g}, \widehat{\boldsymbol{S}}_{g,**}^{\mathrm{UV}}\boldsymbol{R}_{**}\widehat{\boldsymbol{S}}_{g,**}^{\mathrm{UV}}) \qquad (5)$$

$$p(\boldsymbol{\theta}_{.g}|\boldsymbol{\Sigma}_{1:S}, \boldsymbol{\pi}) = \sum_{m=1}^{M}\sum_{s=1}^{S} \pi_{ms}\mathrm{N}(\boldsymbol{\theta}_{.g}|\boldsymbol{0}, \omega_m\boldsymbol{\Sigma}_s), \qquad (6)$$

with Eq (6) unchanged from the fully observed model, i.e., still with $\boldsymbol{\theta}_{.g}$ denoting the full $P$-vector of latent perturbations for gene $g$. When data are MAR, the posterior for $\boldsymbol{\theta}_{.g}$ under model (5)–(6) is a Gaussian mixture available in closed form:

$$p(\boldsymbol{\theta}_{.g}|\widehat{\boldsymbol{\Sigma}}_{1:S}, \widehat{\boldsymbol{\pi}}, \widehat{\boldsymbol{\theta}}_{*g}^{UV}) = \sum_{m=1}^{M}\sum_{s=1}^{S} r_{gms}\mathrm{N}(\boldsymbol{\theta}_{.g}|\boldsymbol{\mu}_{gms}, \boldsymbol{V}_{gms}) \qquad (7)$$

$$\boldsymbol{\mu}_{gms} := \omega_m\widehat{\boldsymbol{\Sigma}}_{s,\cdot *}(\omega_m\widehat{\boldsymbol{\Sigma}}_{s,**} + \widehat{\boldsymbol{S}}_{g,**}^{\mathrm{UV}}\widehat{\boldsymbol{R}}_{**}\widehat{\boldsymbol{S}}_{g,**}^{\mathrm{UV}})^{-1}\widehat{\boldsymbol{\theta}}_{*g}^{\mathrm{UV}}$$

$$\boldsymbol{V}_{gms} := \omega_m\widehat{\boldsymbol{\Sigma}}_{s,\cdot\cdot} - \omega_m\widehat{\boldsymbol{\Sigma}}_{s,\cdot *}(\omega_m\widehat{\boldsymbol{\Sigma}}_{s,**} + \widehat{\boldsymbol{S}}_{g,**}^{\mathrm{UV}}\widehat{\boldsymbol{R}}_{**}\widehat{\boldsymbol{S}}_{g,**}^{\mathrm{UV}})^{-1}\omega_m\widehat{\boldsymbol{\Sigma}}_{s,*\cdot}$$

$$r_{gms} := \frac{\widehat{\pi}_{ms}\mathrm{N}(\widehat{\boldsymbol{\theta}}_{*g}^{\mathrm{UV}}|\boldsymbol{0}, \omega_m\widehat{\boldsymbol{\Sigma}}_{s,**} + \widehat{\boldsymbol{S}}_{g,**}^{\mathrm{UV}}\widehat{\boldsymbol{R}}_{**}\widehat{\boldsymbol{S}}_{g,**}^{\mathrm{UV}})}{\sum_{m,s}\widehat{\pi}_{ms}\mathrm{N}(\widehat{\boldsymbol{\theta}}_{*g}^{\mathrm{UV}}|\boldsymbol{0}, \omega_m\widehat{\boldsymbol{\Sigma}}_{s,**} + \widehat{\boldsymbol{S}}_{g,**}^{\mathrm{UV}}\widehat{\boldsymbol{R}}_{**}\widehat{\boldsymbol{S}}_{g,**}^{\mathrm{UV}})}.$$

### EM algorithm

A detailed derivation of the EM algorithm used to fit the MV model introduced at (2)–(4) is given in S2 Note. Algorithm 1 outlines the computations required where, for notational brevity, we use superscripted assignment notation in place of for loops; for example, $\longleftarrow^{g,m,s}$ performs an assignment for each $(g, m, s) \in \{g = 1\ldots,G\} \times \{m = 1\ldots,M\} \times \{s = 1\ldots,S\}$.

---

**Algorithm 1** EM algorithm targeting $\mathbf{\Sigma}_{1:S}, \boldsymbol{\pi}$.

---

**Inputs:**
 UV model outputs $\widehat{\boldsymbol{\theta}}^{\mathrm{UV}}_{1:G}, \widehat{\boldsymbol{S}}^{\mathrm{UV}}_{1:G}$
 Estimated noise correlation $\widehat{\boldsymbol{R}}$
 Known scale parameters $\omega_{1:M}$
**Initialize:**

$$\pi_{ms} \xleftarrow{m,s} \frac{1}{MS}\,, \ \mathbf{\Sigma}_s \xleftarrow{s} \mathbf{\Sigma}_s^{(0)}$$

**repeat**

$$\boldsymbol{\mu}_{gms} \xleftarrow{g,m,s} \omega_m \mathbf{\Sigma}_{s,\cdot *}(\omega_m \mathbf{\Sigma}_{s,**} + \widehat{\boldsymbol{S}}^{\mathrm{UV}}_{g,**}\widehat{\boldsymbol{R}}_{**}\widehat{\boldsymbol{S}}^{\mathrm{UV}}_{g,**})^{-1}\widehat{\boldsymbol{\theta}}^{\mathrm{UV}}_{*g}$$

$$\boldsymbol{V}_{gms} \xleftarrow{g,m,s} \omega_m \mathbf{\Sigma}_{s,\cdot\cdot} - \omega_m \mathbf{\Sigma}_{s,\cdot *}(\omega_m \mathbf{\Sigma}_{s,**} + \widehat{\boldsymbol{S}}^{\mathrm{UV}}_{g,**}\widehat{\boldsymbol{R}}_{**}\widehat{\boldsymbol{S}}^{\mathrm{UV}}_{g,**})^{-1}\omega_m \mathbf{\Sigma}_{s,*\cdot}$$

$$r_{gms} \xleftarrow{g,m,s} \frac{\widehat{\pi}_{ms}\mathrm{N}(\widehat{\boldsymbol{\theta}}^{\mathrm{UV}}_{*g}|\boldsymbol{0}, \omega_m \mathbf{\Sigma}_{s,**} + \widehat{\boldsymbol{S}}^{\mathrm{UV}}_{g,**}\widehat{\boldsymbol{R}}_{**}\widehat{\boldsymbol{S}}^{\mathrm{UV}}_{g,**})}{\sum\limits_{m,s} \pi_{ms}\mathrm{N}(\widehat{\boldsymbol{\theta}}^{\mathrm{UV}}_{*g}|\boldsymbol{0}, \omega_m \mathbf{\Sigma}_{s,**} + \widehat{\boldsymbol{S}}^{\mathrm{UV}}_{g,**}\widehat{\boldsymbol{R}}_{**}\widehat{\boldsymbol{S}}^{\mathrm{UV}}_{g,**})}$$

$$\pi_{ms} \xleftarrow{m,s} \frac{\sum\limits_{g} r_{gms}}{\sum\limits_{m,s,g} r_{gms}}$$

$$C_s \xleftarrow{s} \frac{\sum\limits_{g,m} r_{gms}(\boldsymbol{V}_{gms} + \boldsymbol{\mu}_{gms}\boldsymbol{\mu}_{gms}^T)/\omega_m}{\sum\limits_{g,m} r_{gms}}$$

$$\mathbf{\Sigma}_s \xleftarrow{s} \underset{\mathbf{\Sigma}\in\mathcal{W}_K}{\mathrm{argmax}} \ \log|\mathbf{\Sigma}| + \mathrm{tr}(\mathbf{\Sigma}^{-1}C_s)$$

 where $\mathcal{W}_K := \{\boldsymbol{W}\boldsymbol{W}^T + \boldsymbol{\Psi}_s : \ \boldsymbol{W} \in \mathbb{R}^{P\times K}, \boldsymbol{\Psi}_s \text{ diagonal with } [\boldsymbol{\Psi}_s]_{jj} \geq 0 \ \forall j\}$
 **until** convergence
**Outputs:** $\widehat{\mathbf{\Sigma}}_{1:S}, \widehat{\boldsymbol{\pi}}$

---

**Initialisation.** The key parameters to initialise are the $S$ covariance matrices $\mathbf{\Sigma}_{1:S}$. We initialise $\mathbf{\Sigma}_s$ at the sample covariance matrix calculated using a specified subset of samples, denoted $\mathcal{J}_s$:

$$\mathbf{\Sigma}_s^{(0)} := \widehat{\mathrm{cov}}_{P\times P}(\widehat{\mathbf{\Theta}}^{\mathrm{UV}}_{\cdot\mathcal{J}_s}) \tag{8}$$

where $\widehat{\mathbf{\Theta}}^{\mathrm{UV}}$ is the $P\times G$ matrix of UV results; any missing data in $\widehat{\mathbf{\Theta}}^{\mathrm{UV}}$ are zero-filled just for the purposes of the above calculation. Should this yield a positive semidefinite $\mathbf{\Sigma}_s^{(0)}$, we add $\varepsilon\boldsymbol{I}$ to ensure positive definiteness at initialisation; for results shown here, we use $\varepsilon = 0.05$.

Our main results are based upon the case of a single covariance matrix ($S = 1$) in which case $\mathcal{J}_1$ comprises all (nonsynthetic null) samples in the training set. When the model is specified to have more than 1 covariance matrix ($S > 1$), we choose the subsets $\mathcal{J}_{1:S}$ to partition the training set via model-based clustering of the zero-completed version of $\widehat{\Theta}^{UV}$ using the function Mclust() in the R package mclust with default parameter settings.

The likelihood of the MV model (2)–(4) is multimodal, and, hence, convergence of the EM algorithm is sensitive to initialisation. This sensitivity is investigated in a number of ways as part of our model checking section in Methods–*Model checking and sensitivity analyses*. By repeating our entire analysis for data subsets, e.g., of size 500, we capture variation in empirical initialisation (since the $\Sigma_s^{(0)}$ are based only on the training data) as well as variation in the likelihood surface from data subsampling; we demonstrate that our results are robust to the combination of these 2 types of variation.

The initialisation at the empirical covariance matrix (8) is helpful for enabling the EM algorithm to target the global optimum. To demonstrate this, we investigate random, vanilla initialisation of $\Sigma$, setting

$$\Sigma^{(0)} \sim \text{Inverse-Wishart}(I, 2000). \tag{9}$$

We perform this random initialisation for 10 data subsets of size 2,000 in the single-covariance matrix case $S = 1$ (these are the same data subsets used for the main cross-validated analysis). Then, in each of these cases, we examine the value of the cross-validated likelihood fit and compare it to the sample covariance initialised cross-validated likelihood fit (S10 Fig, Methods–*Cross-validation and model averaging* and Methods–*Cross-validated likelihood for IMPC data*). Across 10 folds, the randomly initialised fits perform systematically worse in all cases in terms of CV likelihood, illustrating the benefits of using a supervised initialisation in this context to mitigate the nonconvexity of the optimisation.

There are potential enhancements to the EM algorithm to increase the probability of convergence to the global maximum, such as the split and merge algorithm of [23]. While our basic EM implementation appears to provide good performance for the datasets considered here, particularly with reasonable initialisation, it could usefully be extended to incorporate such enhancements in future.

**Convergence.** The EM algorithm is deemed to have converged when the change in objective function between consecutive iterations falls below a tolerance threshold. We choose the tolerance threshold adaptively, with reference to variation in log likelihood contribution across samples. Specifically, denoting the contribution of the $g$th sample to the log likelihood at the $t$th iteration by $l_g^{(t)}$, the tolerance is set to tol:=$\varepsilon_{\text{tol}} N_{\text{tra}} \text{MAD}(\{l_1^{(t)}, \ldots, l_{N_{\text{tra}}}^{(t)}\})$, where MAD() denotes the median absolute deviation, $N_{\text{tra}}$ the number of training samples, and $\varepsilon_{\text{tol}}$ a user-specified constant (we used $\varepsilon_{\text{tol}} = 10^{-4}$).

## Control of error rates

Statistical measures of model fit, such as $p$-values and Bayes factors, are especially useful tools when the true data generating mechanism lies within the space of statistical models hypothesised. When the model space excludes the true mechanism, measures of statistical significance can become miscalibrated [47]. This is particularly important in highly structured scientific data, where incorrectly assuming conditional independence in a model can lead to artificially tight confidence intervals and inflated testing false positive rates.

Our solution to this is to use a nonparametric approach to error rate control, known as the Westfall–Young permutation method [32,48]. The essence of this approach is quite simple—

we generate synthetic null data that mimic the structure of the actual data as precisely as possible but that by design do not deviate systematically from WT animals; in our approach, synthetic null lines are drawn at random from WT samples. These null lines serve as the set of true null hypotheses in our implementation of the Westfall–Young permutation approach to error rate control [32,48].

We refer to a number of different error rates, so in Table 3, we present a glossary relating notation to a brief description and where each error rate is defined. For the main analyses in our paper, we *control* $\widehat{\mathrm{Fdr}}_{\mathrm{complete}}$ using the Westfall–Young permutation approach. We *monitor* $\widehat{\mathrm{Fdr}}_{\mathrm{single}}$ and $\widehat{\mathrm{Fsr}}_{\mathrm{replicate}}$. Finally, we also control lfsr (using Westfall–Young, and also nominally) in Methods–*Comparison with existing methods* for the purposes of benchmarking.

**Synthetic null data.** We define a synthetic null line to be a subsample of typically 10 to 20 WT animals chosen at random so as to reflect the experimental design properties of an actual KO line. Synthetic nulls play important methodological roles in our inference: most importantly in permutation-based control of the Fdr when calling phenotype hits, but also in estimation of the experimental correlation matrix $\boldsymbol{R}$ in the MV model at (2). Synthetic null lines are generated by randomly selecting groups of WT animals from a single centre so as to match the experimental design characteristics of a particular true KO line at that centre. Specifically, for each litter of the true KO line, we sample from a computationally matched WT litter at the same centre. For a KO litter with $l$ animals that was first phenotyped on day $d$, we sample a WT litter from all possible WT litters at the same phenotyping centre having at least $l$ animals and randomly select $l$ animals from that litter. Litters that were measured closer in time to day $d$ are selected with higher probability [11].

**Hypothesis testing and Fdr.** A vital output of the IMPC is the data-driven compilation of a list of (phenotype, KO gene) pairs at which there is evidence for the phenotype being perturbed by the gene KO. This leads us to the analytical goal of testing the null hypothesis $H^0\!: \theta_{pg} = 0$ with high statistical power under a controlled false positive rate. The IMPC data have many levels of complex structure, resulting in potential for model misspecification and inflated false positive rates for parametric tests. Further, the IMPC's massive number of often strongly correlated tests calls for an effective power-preserving multiple testing correction. We address these challenges by controlling the Fdr using the Westfall–Young permutation approach, which provides robustness to model misspecification in combination with high statistical power when tests are correlated [32,48]; the synthetic null lines $\mathcal{S}$ serve as the set of true null hypotheses.

To test the null hypothesis of no perturbation of phenotype $p$ in KO line $g$, we use a $z$-statistic defined as the ratio of posterior mean to posterior SD, i.e.,

$$z_{pg}^{\mathrm{MV}} := \frac{\widehat{\theta}_{pg}^{\mathrm{MV}}}{\widehat{s}_{pg}^{\mathrm{MV}}} \tag{10}$$

**Table 3. Glossary of error rates referred to in this paper.**

| Definition | Notation | Description |
|---|---|---|
| (12) | Fdr | False discovery rate [30] |
| (14) | Fdr$_{\mathrm{single}}$ | Fdr for rejecting *single* null, i.e., null at phenotype–gene pair |
| (15) | $\widehat{\mathrm{Fdr}}_{\mathrm{single}}$ | Estimator for Fdr$_{\mathrm{single}}$ |
| (16) | Fdr$_{\mathrm{complete}}$ | Fdr for rejecting *complete* null, i.e., "all phenotypes null" at a gene |
| (17) | $\widehat{\mathrm{Fdr}}_{\mathrm{complete}}$ | Estimator for Fdr$_{\mathrm{complete}}$ |
| (19) | Fsr | False sign rate [49] |
| (24) | $\widehat{\mathrm{Fsr}}_{\mathrm{replicate}}$ | Estimator for Fsr based on replicated measurements |
| (33) | lfsr | Local false sign rate [27] |

with the corresponding definition for the UV model output. We choose a significance threshold, denoted $\tau$, so that if

$$|z_{pg}| > \tau \qquad (11)$$

then line $g$ is called as significantly perturbed at phenotype $p$, with directionality determined by the sign of $z_{pg}$.

We choose $\tau$ so as to control the Fdr. We use the "Bayesian" Fdr definition [30]:

$$\text{Fdr}(\mathcal{C}) := \mathbb{P}(H^0 \text{true} | T \in \mathcal{C}). \qquad (12)$$

where $H^0$ is a null hypothesis, $T$ a test statistic, and $\mathcal{C}$ a critical (rejection) region, which is chosen to control the corresponding empirical form [30]

$$\overline{\text{Fdr}}(\mathcal{C}) := \frac{\mathbb{P}(H^0 \text{true})\mathbb{P}(T \in \mathcal{C} | H^0 \text{true})}{\frac{1}{N}\sum_{i=1}^{N} \mathbb{I}(T_i \in \mathcal{C})}. \qquad (13)$$

The definition of Fdr in (13) is conservative in the sense that our control of $\overline{\text{Fdr}}(\mathcal{C})$ implies similar control of the Benjamini–Hochberg FDR [30,50]. Our choice of (13) is primarily motivated by convenience: Synthetic null data allow the term $\mathbb{P}(T \in \mathcal{C} | H^0 \text{ true})$ in the numerator of (13) to be estimated and controlled. The other terms in (13) can be dealt with straightforwardly: The denominator is known, and the prior $\mathbb{P}(H^0 \text{ true})$ can be specified, conservatively at 1 as we do here, or informatively when prior information is available. We estimate Fdr at 2 levels of granularity: the phenotype–gene pair, and the gene. At the phenotype–gene pair level (i.e., for a *single* test), $\text{Fdr}_{\text{single}}$ is the Fdr resulting from rejecting the null hypothesis, $H^0_{pg} : \theta_{pg} = 0$, at each phenotype $p$, gene $g$ pair for which $|z_{pg}| \geq \tau$:

$$\text{Fdr}_{\text{single}}(\tau) := \mathbb{P}(\theta_{pg} = 0 | |z_{pg}| \geq \tau). \qquad (14)$$

We estimate $\text{Fdr}_{\text{single}}(\tau)$ by

$$\widehat{\text{Fdr}}_{\text{single}}(\tau) := \frac{\mathbb{P}(H^0_{pg} \text{ true})\frac{1}{|\mathcal{S}|}\sum_{p}\sum_{g \in \mathcal{S}}\mathbb{I}[|z_{pg}| \geq \tau]}{\frac{1}{|\mathcal{K}|}\sum_{p}\sum_{g \in \mathcal{K}}\mathbb{I}[|z_{pg}| \geq \tau]} \qquad (15)$$

where $z_{pg}$ from synthetic null lines are included in the numerator to estimate the second term in the numerator of (13).

At the gene level, $\text{Fdr}_{\text{complete}}$ is the Fdr resulting from rejecting the *complete* null hypothesis $H^0_g : \theta_{pg} \equiv 0 \; \forall p$ across all phenotypes at each gene for which $\max_p\{|z_{pg}|\} \geq \tau$,

$$\text{Fdr}_{\text{complete}}(\tau) := \mathbb{P}(\theta_{pg} \equiv 0 \; \forall p | \max_p\{|z_{pg}|\} \geq \tau), \qquad (16)$$

with its empirical form $\overline{\text{Fdr}}_{\text{complete}}(\tau)$ defined analogously to (13). We estimate $\overline{\text{Fdr}}_{\text{complete}}(\tau)$ by

$$\widehat{\text{Fdr}}_{\text{complete}}(\tau) := \frac{\mathbb{P}(H^0_g \text{ true})\frac{1}{|\mathcal{S}|}\sum_{g \in \mathcal{S}}\mathbb{I}\left[\max_p\{|z_{pg}|\} \geq \tau\right]}{\frac{1}{|\mathcal{K}|}\sum_{g \in \mathcal{K}}\mathbb{I}\left[\max_p\{|z_{pg}|\} \geq \tau\right]} \qquad (17)$$

where again $z_{pg}$ from synthetic null lines are included in the numerator.

We monitor both $\widehat{\text{Fdr}}_{\text{single}}$ and $\widehat{\text{Fdr}}_{\text{complete}}$ as related but distinct estimates of Fdr. However, it is $\widehat{\text{Fdr}}_{\text{complete}}$ that we use to select $\tau$ to control $\overline{\text{Fdr}}_{\text{complete}}(\tau) \leq \alpha$ while maximising power, via the

optimization

$$\tau(\alpha) = \mathrm{argmin}_\tau \tau \text{ satisfying } \widehat{\mathrm{Fdr}}_{\mathrm{complete}}(\tau) \leq \alpha. \tag{18}$$

Control of $\overline{\mathrm{Fdr}}_{\mathrm{complete}}$ via the permutation based (i.e., synthetic null based) $\widehat{\mathrm{Fdr}}_{\mathrm{complete}}$ is an implementation of the Westfall–Young permutation procedure [31,32,48].

**Replicability and false sign rates.** When a KO line is phenotyped in multiple laboratories, calling hits (identifying significant perturbations) in the same direction in both laboratories is supportive of a method's replicability. In Fig 6, concordant hits correspond to points in the blue shaded regions. In contrast, hits presenting an increasing phenotype in one laboratory and a decreasing one in the other imply that at least one of the 2 hits is a false positive (indicated by the red regions in Fig 6). In our analyses, we examine such concordance across pairs of contexts (across laboratories as just introduced, and also to compare heterozygotes versus homozygotes). It is useful to be able to relate the degree of observed replicability to an underlying error rate, as this provides extra validation of effective error rate control. We therefore develop a method for quantifying (dis)agreement: a replicability-based estimate of the Fsr, which we denote $\widehat{\mathrm{Fsr}}_{\mathrm{replicate}}$ and derive below. The method maps a contingency table of signed annotations to a compatible Fsr.

The general situation of interest has pairs of signed significance calls outputted in 2 conditionally independent contexts. We represent calls in this section by $\{-1,0,1\}$ with 1 and $-1$ denoting significant positive or negative phenotypic perturbations, $z > \tau$ and $z < -\tau$, respectively, and zero denoting the nonsignificant result $|z| < \tau$. The general set of concordance data can be represented by counts as in Table 4 where the number of points in the blue and red regions of Fig 6 are denoted by $n_{--} + n_{++}$ and $n_{+-} + n_{-+}$, respectively.

We will show that, while Table 4 provides little information about Fdr, the probabilities underlying Table 4 can be usefully related to the Fsr, defined as the probability of incorrectly estimating the sign of an effect (making a "type S error") given that the null hypothesis of zero effect is rejected [49]:

$$\mathrm{Fsr} := \mathbb{P}(\mathrm{sign}(z) \neq \mathrm{sign}(\theta) || z| > \tau). \tag{19}$$

We motivate our derivation of an estimator for Fsr by considering the following ratio, $\widehat{q}$, which increases with the level of discordance in Table 4:

$$\widehat{q} := \frac{n_{+-} + n_{-+}}{n_{+-} + n_{-+} + n_{--} + n_{++}}. \tag{20}$$

We note that

$$\mathbb{E}\,\widehat{q} = q := \mathrm{P}(A, B \text{ disagree}|\text{both } A \text{ and } B \text{ significant})$$

where $A, B \in \{-1,0,1\}$, and we express

$$q \equiv \mathrm{P}(A \neq B | AB \neq 0)$$

$$= \mathrm{P}(A \neq B | AB \neq 0, \theta = 0)\mathbb{P}(\theta = 0 | AB \neq 0)$$

$$+ \mathrm{P}(A \neq B | AB \neq 0, \theta \neq 0)\mathbb{P}(\theta \neq 0 | AB \neq 0) \tag{21}$$

$$= \frac{1}{2}\psi + 2\,\mathrm{Fsr}(1 - \mathrm{Fsr})(1 - \psi) \tag{22}$$

in which we have defined $\psi := \mathbb{P}(\theta = 0 | AB \neq 0)$, where $\psi$ is interpretable as a "double false discovery rate," i.e., the probability of the null ($\theta = 0$) being true given it has been rejected in 2 conditionally independent tests, e.g., on data sets gathered in 2 different laboratories (note that $\psi = O(\text{Fdr}^2)$ is small under reasonable control of Fdr). Further, in the step from (21) to (22) we have assumed

$$P(A \neq B | AB \neq 0, \theta = 0) = \frac{1}{2},$$

i.e., that false positives are equally likely to be in the positive or negative direction. We also used the following in the step from (21) to (22):

$$P(A \neq B | AB \neq 0, \theta \neq 0) = P([A \text{ false sign and } B \text{ true sign}] \vee [B \text{ false sign and } A \text{ true sign}])$$

$$= 2 \, \text{Fsr}(1 - \text{Fsr}).$$

Solving (22) for Fsr gives:

$$\text{Fsr} = \frac{1}{2}\left(1 - \sqrt{1 - \frac{2q - \psi}{1 - \psi}}\right). \tag{23}$$

The right-hand side of (23) is a decreasing function of $\psi$, so we define a conservative (slightly upwardly biased) estimator of Fsr by setting $\psi = 0$:

$$\widehat{\text{Fsr}}_{\text{replicate}} := \frac{1}{2}\left(1 - \sqrt{1 - 2\widehat{q}}\right) \tag{24}$$

where $q$ has been replaced by the estimator $\widehat{q}$ defined at (20). We obtain an approximate confidence interval for Fsr by substituting (in place of $\widehat{q}$ in (24)) exact binomial confidence interval bounds for $q$ derived under a model where the number of disagreements follows a binomial distribution with success probability $q$:

$$n_{+-} + n_{-+} \sim \text{Binomial}(n_{+-} + n_{-+} + n_{--} + n_{++}, q).$$

**Cross-validation and model averaging.** Highly parameterised statistical models can overfit data, resulting in poor out-of-sample performance. This overfitting concern applies to the MV model here, as it has a flexible and high-dimensional covariance matrix parameterisation, which is learned empirically, although it is somewhat mitigated by the structural regularisation via a factor model representation at (4). To protect against overfitting, all MV results are inferred within a cross-validation framework whereby we split the data set into "training" and "test" sets $C$ times, and then combine test set results across splits using Bayesian model averaging.

**Table 4. Replicability table between methods A and B.**

|   |   |   | B |   |   |
|---|---|---|---|---|---|
|   |   | −1 | 0 | 1 |
|   | 1 | $n_{+-}$ | $n_{+0}$ | $n_{++}$ |
| A | 0 | $n_{0-}$ | $n_{00}$ | $n_{0+}$ |
|   | −1 | $n_{--}$ | $n_{-0}$ | $n_{-+}$ |

We denote each line $g$ as either being in the set $\mathcal{K}$ of true KO lines, or in the set $\mathcal{S}$ of synthetic null lines; each of these 2 sets comprises $N_{\text{tot}} = 4{,}548$ lines (since each synthetic null line matches the design of a true KO line).

For each cross-validation split $c$ of $1,\ldots,C$, we randomly partition $\mathcal{K}$ into a training set $\mathcal{K}_{\text{tra}}^{(c)}$ of size $N_{\text{tra}}$ and a test set $\mathcal{K}_{\text{tes}}^{(c)}$ of size $N_{\text{tot}}-N_{\text{tra}}$. We randomly partition $\mathcal{S}$ similarly and independently into $\mathcal{S}_{\text{tra}}^{(c)}$ and $\mathcal{S}_{\text{tes}}^{(c)}$. We proceed to estimate $\boldsymbol{\Sigma}_{1:S}^{(c)}, \boldsymbol{\pi}^{(c)}, \boldsymbol{R}^{(c)}$ using training genes, i.e., $g \in \mathcal{K}_{\text{tra}}^{(c)} \cup \mathcal{S}_{\text{tra}}^{(c)}$. We then estimate $\boldsymbol{\theta}_{\cdot g}^{(c)}$ conditional on $\widehat{\boldsymbol{\Sigma}}_{1:S}^{(c)}, \widehat{\boldsymbol{\pi}}^{(c)}, \widehat{\boldsymbol{R}}^{(c)}$ using test genes, i.e., $g \in \mathcal{K}_{\text{tes}}^{(c)} \cup \mathcal{S}_{\text{tes}}^{(c)}$. Test set estimates of $\boldsymbol{\theta}_{\cdot g}^{(c)}$ are combined across cross-validation splits using Bayesian model averaging, i.e.,

$$p(\boldsymbol{\theta}_{\cdot g}|\boldsymbol{\theta}_{\cdot g}^{\text{UV}}) = \frac{\displaystyle\sum_{c \in \{c : g \in \mathcal{K}_{\text{tes}}^{(c)}\}} p(\boldsymbol{\theta}_{\cdot g}|\boldsymbol{\theta}_{\cdot g}^{\text{UV}}, \widehat{\mathcal{P}}^{(c)}) p(\widehat{\boldsymbol{\theta}}_{\cdot g}^{\text{UV}}|\widehat{\mathcal{P}}^{(c)})}{\displaystyle\sum_{c \in \{c : g \in \mathcal{K}_{\text{tes}}^{(c)}\}} p(\widehat{\boldsymbol{\theta}}_{\cdot g}^{\text{UV}}|\widehat{\mathcal{P}}^{(c)})} \tag{25}$$

$$\widehat{\mathcal{P}}^{(c)} := \{\widehat{\boldsymbol{R}}^{(c)}, \widehat{\boldsymbol{\Sigma}}_{1:S}^{(c)}, \widehat{\boldsymbol{\pi}}^{(c)}\},$$

which represents the combined posterior $p(\boldsymbol{\theta}_{\cdot g}|\widehat{\boldsymbol{\theta}}_{\cdot g}^{\text{UV}})$ as a mixture of the split-specific posteriors, $p(\boldsymbol{\theta}_{\cdot g}|\widehat{\boldsymbol{\theta}}_{\cdot g}^{\text{UV}}, \widehat{\mathcal{P}}^{(c)})$, each of which is a Gaussian mixture. We define posterior MV estimates $\widehat{\boldsymbol{\theta}}_{\cdot g}^{\text{MV}}, \widehat{\boldsymbol{s}}_{\cdot g}^{\text{MV}}$ as the mean and standard deviation of the combined posterior $p(\boldsymbol{\theta}_{\cdot g}|\widehat{\boldsymbol{\theta}}_{\cdot g}^{\text{UV}})$ in (25), and these estimates are taken forward to phenotype calling under a controlled Fdr. Our framework for cross-validated empirical Bayes inference is laid out in Table 5.

Additionally, we calculate a pooled covariance estimate $\widehat{\boldsymbol{\Sigma}}_{\text{pooled}}$ by Bayesian model averaging across the $C$ split-specific models using test set data, i.e.,

$$\widehat{\boldsymbol{\Sigma}}_{\text{pooled}} = \frac{\displaystyle\sum_{c=1}^{C} \widehat{\boldsymbol{\Sigma}}^{(c)} \prod_{g \in \mathcal{K}_{\text{tes}}^{(c)}} p(\widehat{\boldsymbol{\theta}}_{\cdot g}^{\text{UV}}|\widehat{\mathcal{P}}^{(c)})}{\displaystyle\sum_{c=1}^{C} \prod_{g \in \mathcal{K}_{\text{tes}}^{(c)}} p(\widehat{\boldsymbol{\theta}}_{\cdot g}^{\text{UV}}|\widehat{\mathcal{P}}^{(c)})} \tag{26}$$

$$\widehat{\boldsymbol{\Sigma}}^{(c)} := \sum_{m,s} \widehat{\pi}_{ms}^{(c)} \omega_m \widehat{\boldsymbol{\Sigma}}_s^{(c)},$$

taking forward $\widehat{\boldsymbol{\Sigma}}_{\text{pooled}}$ from (26) to factor analysis (Methods–*Factor model*).

## Model checking and sensitivity analyses

**Sensitivity analysis.** We verify that our downstream factor analysis of $\widehat{\boldsymbol{\Sigma}}$ is robust by comparing results across different cross-validation folds. Specifically, we compare the varimax-rotated factor loadings from our final estimate $\widehat{\boldsymbol{\Sigma}}_{\text{pooled}}$ defined at (26) to those from each fold $c$ and select the $\widehat{\boldsymbol{\Sigma}}^{(c)}$ showing highest discrepancy based on the symmetrized KL divergence $(\widetilde{D}_{\text{KL}}(P||Q) := D_{\text{KL}}(P||Q) + D_{\text{KL}}(Q||P))$,

$$c' = \underset{c}{\arg\max} \; \widetilde{D}_{\text{KL}}(\text{N}(0, \widehat{\boldsymbol{\Sigma}}_{\text{pooled}})||\text{N}(0, \widehat{\boldsymbol{\Sigma}}^{(c)})). \tag{27}$$

The loadings plots for $\widehat{\boldsymbol{\Sigma}}_{\text{pooled}}$ and $\widehat{\boldsymbol{\Sigma}}^{(c')}$ are compared in S11 Fig and appear qualitatively similar

with differences only at a small number of factors. These limited differences are due to merging or splitting of factors across the 2 decompositions. Our conclusion here is that the factor analysis is relatively insensitive to data subsampling: Variation across factor decompositions should occur at only a small number of factors in the worst case scenario.

**Data subsampling.** Here, we examine the stability of results to potential MV heteroscedasticity across KO lines that may not be captured by our MV mixture model. We perform a sensitivity analysis in which we subsample 500 of the total 4,548 lines at random and use them as the training set to refit the MV model. We perform this subsampling $c = 1,\ldots,10$ times, each time estimating $\widehat{\mathbf{\Sigma}}^{(c)}$ from the training set and retaining the MV model phenotype calls from the test set. We find the fold, $c'$, with the greatest symmetrized KL divergence between $N(\mathbf{0}, \widehat{\mathbf{\Sigma}}^{(c)})$ to $N(\mathbf{0}, \widehat{\mathbf{\Sigma}}_{\mathrm{pooled}})$ as at (27).

In Table 6, we compare fold $c'$'s signed phenotype calls to the corresponding calls in the full analysis (i.e., the cross-validated and model-combined analysis at (26)). The level of discordance is low with a total of 37 disagreements across 8,316 instances where both models call a hit. (Note that we cannot estimate Fsr effectively from Table 6, because of the conditional dependence between test results from the full and subsampled analyses.) It is a reassuring quality control check that picking the most discrepant subsample of size 500 leads only to this small level of discordance, suggesting that a reduced sample size, while reducing power, should not lead to any qualitative disagreement with the conclusions of the full-data analysis.

**Predicting masked data.** It can be seen from Fig 2 that data are often missing across an entire procedure for a KO line. To check missing data inference, we perform the following "mask-predict-compare" algorithm: (i) for each KO line in the test set, artificially mask data from each of its measured procedures in turn; (ii) predict the perturbations underlying the masked data; and (iii) compare the predicted perturbations to those estimated by the UV model on the unmasked data. We refer to the inference on masked data as leave-one-procedure-out MV (LOO-MV). The level of discordance between the LOO-MV and UV results is low (S9 Fig) and is compatible with $\widehat{\mathrm{Fsr}}_{\mathrm{replicate}} = 0.4\%$ (95% CI: 0.2% to 0.7%). This is consistent with the false positive rate being well calibrated even when inference is performed in the presence of missing data.

**Examining discordance between the MV model and the IMPC database.** Referring back to Results–*Comparison with IMPC database* and Table 1(B), here, we inspect these 3 cases of disagreement more closely by examining which directionality (our MV model or the IMPC database) is more biologically sensible. We use empirical Bayes to quantify prior beliefs about hit directionality at any particular phenotype $p$ as a probability: $\mathbb{P}_{\mathrm{prior}}(\theta_{pg} > 0 | \theta_{pg} \neq 0)$. This involves aggregating information on directionality from that phenotype's hits across all genes, which can be done via a simple average:

$$\mathbb{P}_{\mathrm{prior}}\left(\theta_{pg} > 0 | \theta_{pg} \neq 0\right) := \frac{\sum_g \mathbb{I}(\widehat{\theta}_{pg} > 0 | \mathrm{call}\ \theta_{pg} \neq 0)}{\sum_g \mathbb{I}(\mathrm{call}\ \theta_{pg} \neq 0)}. \tag{28}$$

Since there is no disagreement between our UV model's calls and those in the IMPC database, we include hits from both these methods in (28), but we do not include calls from our MV model. We use the prior defined in (28) to analyse the 3 signed phenotype hits showing disagreement between our MV model and the IMPC database, yielding a Bayes factor of 1.45 supportive of the MV model, but this is a weak Bayes factor that does not provide substantial evidence either way on which model's outputted directionality is most sensible on these instances of disagreement; this negative result makes sense given the small sample size of 3 leading to low power.

**Table 5. Cross-validated empirical Bayes inference.**

| Stage | Input | Output | Samples | Method |
|---|---|---|---|---|
| UV model | Raw data $Y$ | $\widehat{\boldsymbol{\theta}}_{\cdot g}^{\text{UV}}, \widehat{\boldsymbol{s}}_{\cdot g}^{\text{UV}}$ | $\mathcal{K}, \mathcal{S}$ | Hierarchical Bayes (MCMC) |
| Train (split $c$) | $\widehat{\boldsymbol{\theta}}_{\cdot g}^{\text{UV}}, \widehat{\boldsymbol{s}}_{\cdot g}^{\text{UV}}$ | $\widehat{\boldsymbol{R}}^{(c)}$ | $\mathcal{S}_{\text{tra}}^{(c)}$ | Weighted sample correlation |
| Train (split $c$) | $\widehat{\boldsymbol{\theta}}_{\cdot g}^{\text{UV}}, \widehat{\boldsymbol{s}}_{\cdot g}^{\text{UV}}, \widehat{\boldsymbol{R}}^{(c)}$ | $\widehat{\boldsymbol{\Sigma}}_{1:S}^{(c)}, \widehat{\boldsymbol{\pi}}^{(c)}$ | $\mathcal{K}_{\text{tra}}^{(c)}$ | MAP estimation (EM) |
| Fit (split $c$) | $\widehat{\boldsymbol{\theta}}_{\cdot g}^{\text{UV}}, \widehat{\boldsymbol{s}}_{\cdot g}^{\text{UV}}, \widehat{\mathcal{P}}^{(c)}$ | $\widehat{\boldsymbol{\theta}}_{\cdot g}^{(c)}, \widehat{\boldsymbol{s}}_{\cdot g}^{(c)}$ | $\mathcal{K}_{\text{tes}}^{(c)}, \mathcal{S}_{\text{tes}}^{(c)}$ | Conjugate Bayesian inference |
| Combine splits | $\widehat{\boldsymbol{\theta}}_{\cdot g}^{(1:C)}, \widehat{\boldsymbol{s}}_{\cdot g}^{(1:C)}, \widehat{\mathcal{P}}^{(1:C)}$ | $\widehat{\boldsymbol{\theta}}_{\cdot g}^{\text{MV}}, \widehat{\boldsymbol{s}}_{\cdot g}^{\text{MV}}$ | $\mathcal{K}, \mathcal{S}$ | Bayesian model averaging |
| Hit calling | $\widehat{\boldsymbol{\theta}}_{\cdot g}^{\text{MV}}, \widehat{\boldsymbol{s}}_{\cdot g}^{\text{MV}}$ | $\widehat{\mathbb{I}}(\theta_{pg} \neq 0)$ | $\mathcal{K}, \mathcal{S}$ | Permutation-controlled Fdr |

EM, expectation–maximisation; Fdr, false discovery rate; MAP, maximum a posteriori; MCMC, Markov chain Monte Carlo; UV, univariate.

## Methods for biological applications

**Gene ontology analysis.** We use the R package GOfuncR to test for co-enrichment between GO terms and IMPC phenotypes. An important feature of this package is that it corrects for multiple testing and interdependency of the tests, using random permutations of the gene-associated variables to control family-wise error rates. We create IMPC–phenotype gene sets comprising genes that are not only significantly perturbed, but also exhibit an effect size of at least 2 times the SD of effect sizes across all genes; in the mathematical notation we have introduced, our IMPC gene set for phenotype $p$ is defined as:

$$\text{IMPC phenotype } p\text{'s gene set} = \{g : |z_{pg}| > \tau \wedge |\widehat{\theta}_{pg}^{\text{MV}}| > 2 \times \text{SD}_p\} \tag{29}$$

$$\text{SD}_p := \text{Sample SD of } \{\widehat{\theta}_{pg}^{\text{MV}} : g = 1, \ldots, G\}. \tag{30}$$

We apply an additional filter to focus only on homozygous KOs. We use a family-wise error rate threshold of 5% (the probability of one or more false positives when testing a single IMPC gene set for co-enrichment against all BP GO terms is constrained to be less than 5%).

We perform 1,000 permutations of the GO graph for each IMPC phenotype. The background gene set (also known as the gene universe) is defined for the MV model to be all homozygote–KO genes at which some phenotype measurements are available (a total of 2,628 genes); for the UV model, the background gene set comprises all homozygote–KO genes at which this particular phenotype is available. The basic inferential tool is a Fisher exact test for independence of row and column classifications in a 2-by-2 contingency table, such as those shown in Fig 7. Implementing this basic test within GOfuncR ensures that error rates, here family-wise error rates, are controlled appropriately.

**Table 6. Comparison of signed phenotype hits for the MV model applied to the most KL-divergent subsampled data set of training size $N = 500$ (left) compared to the full data set of training size $N = 2000$ (top).** We represent calls by a number in $\{-1, 0, 1\}$, with 1 and $-1$ denoting significant positive and negative phenotypic perturbations, respectively, and 0 denoting a lack of statistical significance.

| | −1 | 0 | 1 |
|---|---|---|---|
| −1 | 5,001 | 1,263 | 19 |
| 0 | 14,109 | 564,001 | 10,371 |
| 1 | 18 | 1,044 | 3,278 |

**Factor model.** In order to facilitate interpretation of phenotypic perturbations, we calculate the eigendecomposition of the correlation matrix underlying $\widehat{\boldsymbol{\Sigma}}_{\text{pooled}}$, i.e.,

$$\boldsymbol{D}_{\Sigma}^{-\frac{1}{2}}\widehat{\boldsymbol{\Sigma}}_{\text{pooled}}\boldsymbol{D}_{\Sigma}^{-\frac{1}{2}} = \boldsymbol{Q}\boldsymbol{\Delta}\boldsymbol{Q}^{T} \tag{31}$$

in the varimax() function with default parameters in R [51].

Denoting the rotated sparse loadings $P$-vectors by $\boldsymbol{\lambda}_l$, $l = 1,\ldots,20$, the $l$th factor score for the $g$th KO gene is $u_{lg} := \boldsymbol{\lambda}_l^{T}\boldsymbol{\theta}_{\cdot g}$.

Hypothesis testing is performed to identify significant perturbations in factor scores. Denoting $\widehat{u}_{lg}^{\text{MV}} := \boldsymbol{\lambda}_l^{T}\boldsymbol{m}_{\cdot g}$ and $\widehat{s}_{lg}^{\text{MV}} := \sqrt{\boldsymbol{\lambda}_l^{T}\boldsymbol{V}_g\boldsymbol{\lambda}_l}$, we form test statistics

$$z_{lg}^{\text{MV}} := \frac{\widehat{u}_{lg}^{\text{MV}}}{\widehat{s}_{lg}^{\text{MV}}} \tag{32}$$

and control Fdr analogously to Methods–*Control of error rates*, where instead of phenotypes $p = 1,\ldots,P$ we now have factor scores $l = 1,\ldots,20$.

## Comparison with existing methods

**Extreme deconvolution (XD).** The model used in [23] and which underlies the accompanying software package Extreme Deconvolution (XD) is similar to the likelihood we use (2)–(3) but has a few differences. XD has the constraint $M = 1$, i.e., does not have the multiple scaling parameters $\omega_{1:M}$ introduced in [22]. XD generalises to underlying mixture components with nonzero means $\boldsymbol{\mu}_{1:S}$ that are themselves estimated, i.e., $N(\boldsymbol{\theta}_{\cdot g}|\boldsymbol{\mu}_s, \boldsymbol{\Sigma}_s)$. For the purposes of method comparison, XD is run with $\boldsymbol{\mu}_s \equiv 0$, as the zero-mean model is appropriate for the data sets we analyse here.

XD uses a similar EM algorithm to ours to maximise the likelihood jointly with respect to $\boldsymbol{\Sigma}_{1:S}$ and $\boldsymbol{\pi}$ (and $\boldsymbol{\mu}_{1:S}$ more generally). Throughout the XD EM algorithm optimisation, the rank of each $\boldsymbol{\Sigma}_s$ remains the same as the rank of its initialised value [22]. We initialise XD at the same values $\boldsymbol{\Sigma}_{1:S}$ and $\boldsymbol{\pi}$ as we initialise our own model (except when we are running XD to generate data-driven matrices for mash, in which case we follow the directions in [22]). In summary, any differences between the fit of our model and the fit of XD are driven primarily by the absence of scaling parameters $\omega_{1:M}$ in XD and our factor-model regularisation of $\boldsymbol{\Sigma}_{1:S}$.

**Multivariate adaptive shrinkage (mash).** Our method's model likelihood (2)–(3) is the same as was introduced in [22] and which is the basis for the software package mash. A particularly important insight in [22] is the introduction of a multiscale mixture across a ladder of scales denoted by $\omega_{1:M}$. The authors note the utility of this approach in the context of multi-tissue eQTLs, and we find it also to be useful for MV mouse phenotyping data. We believe the multiscale covariance model form of [22] has the potential to enhance MV inference across a broad range of scientific disciplines.

The key difference between our approach and mash is in how the covariance matrices $\boldsymbol{\Sigma}_{1:S}$ are defined and estimated. We parameterise our model with a small number of regularised covariance matrices (we consider $S = 1,2$ here) and optimise $\boldsymbol{\Sigma}_{1:S}$ and $\boldsymbol{\pi}$ collectively as part of model fitting. In contrast, mash generates and fixes a larger number ($S = P+10$) of covariance matrices $\boldsymbol{\Sigma}_{1:S}$, in advance of optimising (2)–(3) with respect to $\boldsymbol{\pi}$.

In more detail, mash generates 2 distinct types of covariance matrices: 8 data-driven and $P +2$ canonical. The 8 data-driven covariance matrices inputted into mash are a low-rank representation of the empirical phenotypic covariance among those samples exhibiting largest phenotypic effects. Three of the data-driven covariance matrices are generated using the XD

software [23]. In addition, $P+2$ canonical $P \times P$ covariance matrices are generated, comprising the identity matrix, a matrix of ones, and $\boldsymbol{e}_p \boldsymbol{e}_p^T$ for $p = 1,\ldots,P$ where $\boldsymbol{e}_p$ is a $P$-vector with zeros everywhere except for the $p$th element which is set to 1.

While an elegant aspect of mash is that the optimisation with respect to $\boldsymbol{\pi}$ given $\boldsymbol{\Sigma}_{1:S}$ is convex, its generation of covariance matrices involves nonconvex optimisation within the XD software, so there is potentially some sensitivity to initialisation [22]. Our EM algorithm's MAP optimisation with respect to $\boldsymbol{\Sigma}_{1:S}$ and $\boldsymbol{\pi}$ is nonconvex, and we investigate sensitivity to initialisation in Methods–*Initialisation*.

**Hit rates, error rates, and model fit.** We compare the power (hit rates) and error rate (estimated Fdr or Fsr) of the different methods on the IMPC data. We consider a number of hypothesis testing frameworks, determined by their test statistics and critical regions (rejection criteria).

Test statistics for testing for a perturbation of phenotype $p$ at gene $g$:

A. the $z$ statistic $z_{pg} = \widehat{\theta}_{pg}^{\mathrm{MV}} / \widehat{s}_{pg}^{\mathrm{MV}}$

B. the local false sign rate [27]:

$$\mathrm{lfsr}_{pg} := \min\{\mathbb{P}(\theta_{pg} \geq 0 | \widehat{\boldsymbol{\theta}}_{\cdot g}^{\mathrm{UV}}),\ \mathbb{P}(\theta_{pg} \leq 0 | \widehat{\boldsymbol{\theta}}_{\cdot g}^{\mathrm{UV}})\}, \tag{33}$$

intuitively "the probability that we would incorrectly predict the sign of the effect if we were to use our best guess of the sign (positive or negative)" [22].

Critical regions:

1. controlling $\mathrm{Fdr}_{\mathrm{complete}} < 5\%$ via a permutation-based test-statistic threshold $\tau$, i.e., with critical region either $|z_{pg}| > \tau$ or $\mathrm{lfsr}_{pg} < \tau$ (Methods–*Control of error rates* and [32,48]);

2. nominally controlling the local Fsr $\mathrm{lfsr}_{pg} < 5\%$ [22].

Table 7 shows hit rates and error rates under 3 different methods of error rate control, with the subtables corresponding to the test statistics and critical regions defined above: Table 7A presents A1 (test statistic A with critical region 1); Table 7B presents B1; and Table 7C presents B2. Hit rates are shown stratified according to whether the raw data are measured or missing (with 95% nonparametric bootstrap CIs). We display error rate estimates $\widehat{\mathrm{Fdr}}_{\mathrm{complete}}$, $\widehat{\mathrm{Fdr}}_{\mathrm{single}}$, and $\widehat{\mathrm{Fsr}}_{\mathrm{replicate}}$ as defined in Methods–*Hypothesis testing and Fdr*.

**Cross-validated likelihood for IMPC data.** Within the inferential framework described in Methods–*Cross-validation and model averaging*, we calculate the likelihood of the test set data under a model fitted using only the training data. With reference to (2)–(3), the per-sample log cross-validated likelihood for fold $c$ is

$$\mathcal{L}_{\mathrm{CV}}^{(c)} := \frac{1}{N_{\mathrm{tes}}} \sum_{g \in \mathcal{K}_{\mathrm{tes}}^{(c)}} \log \mathrm{MVNormal}(\widehat{\boldsymbol{\theta}}_{\cdot g}^{\mathrm{UV}} | \boldsymbol{0}, \widehat{\boldsymbol{S}}_g^{\mathrm{UV}} \widehat{\boldsymbol{R}} \widehat{\boldsymbol{S}}_g^{\mathrm{UV}} + \sum_{m,s} \omega_m \widehat{\boldsymbol{\Sigma}}_s) \tag{34}$$

where $\mathcal{K}_{\mathrm{tes}}^{(c)}$ is the set of KO genes in the test set of fold $c$ (Methods–*Cross-validation and model averaging*). Table 8 shows benchmarking results where we present the mean of $\mathcal{L}_{\mathrm{CV}}^{(c)}$ ($\pm 2 \times \mathrm{SEM}$) taken across 10 cross-validation folds.

**Cross-validated likelihood on additional data set.** We compare the various MV methods on an additional data set to examine if the same qualitative performance of the various methods persists. A natural data set to use is the multi-tissue eQTL study of Urbut and colleagues [22] on which mash was first developed. The data set comprises 16,069 samples, each of which

**Table 7. Hit rates and error rates compared across models.** The row showing the model and error rate control used in the main analyses in the paper, $\mathcal{M}_{main}$, is highlighted in bold. The highest hit rates are underlined.

(a) Controlling $Fdr_{complete} \leq 5\%$ using $z$ statistic

| Method | S | K | Hit rate in % when data are | | Estimated error rate in % (95% CI) | | |
| --- | --- | --- | --- | --- | --- | --- | --- |
| | | | measured | missing | $\widehat{Fdr}_{complete}$ | $\widehat{Fdr}_{single}$ | $\widehat{Fsr}_{replicate}$ |
| UV | | | 1.4 (1.3, 1.5) | | 5.0 (3.8, 6.5) | 2.3 (1.6, 2.9) | 0.0 (0.0, 3.6) |
| XD | 1 | | 2.0 (1.8, 2.1) | 0.2 (0.2, 0.3) | 5.1 (3.9, 6.4) | 1.7 (1.4, 2.2) | 0.0 (0.0, 2.2) |
| XD | 2 | | 2.4 (2.2, 2.6) | 0.4 (0.3, 0.4) | 5.0 (3.9, 6.3) | 1.6 (1.2, 2.0) | 0.0 (0.0, 1.8) |
| mash | 158 | | 0.4 (0.3, 0.4) | 0.0 (0.0, 0.1) | 5.1 (2.9, 8.2) | 1.3 (0.8, 1.8) | 0.0 (0.0, 8.1) |
| ComposeMV | 1 | 15 | 10.1 (9.8, 10.5) | 1.9 (1.8, 2.0) | 4.9 (4.1, 5.8) | 1.4 (1.2, 1.7) | 2.6 (1.6, 4.1) |
| **ComposeMV** | **1** | **20** | **<u>10.5 (10.2, 10.9)</u>** | **1.3 (1.2, 1.4)** | **5.0 (4.2, 5.8)** | **1.5 (1.3, 1.8)** | **1.2 (0.6, 2.4)** |
| ComposeMV | 1 | 30 | 9.7 (9.4, 10.0) | 1.1 (1.1, 1.2) | 5.0 (4.2, 5.9) | 1.6 (1.3, 1.9) | 1.0 (0.4, 2.4) |
| ComposeMV | 1 | 40 | 9.4 (9.1, 9.7) | 1.0 (1.0, 1.1) | 5.0 (4.2, 5.9) | 1.6 (1.3, 1.9) | 1.3 (0.6, 2.7) |
| ComposeMV | 2 | 15 | 9.4 (9.0, 9.7) | <u>2.0 (1.8, 2.1)</u> | 5.0 (4.2, 5.9) | 2.2 (1.7, 2.8) | 2.4 (1.4, 4.2) |
| ComposeMV | 2 | 20 | 8.6 (8.2, 8.9) | 1.6 (1.5, 1.8) | 5.0 (4.1, 5.9) | 2.1 (1.6, 2.5) | 1.6 (0.8, 3.1) |
| ComposeMV | 2 | 30 | 8.9 (8.6, 9.2) | 1.4 (1.3, 1.5) | 4.9 (4.1, 5.9) | 2.0 (1.5, 2.5) | 1.1 (0.5, 2.5) |
| ComposeMV | 2 | 40 | 9.3 (9.0, 9.6) | 1.3 (1.2, 1.4) | 5.0 (4.2, 5.9) | 2.1 (1.6, 2.6) | 1.1 (0.5, 2.6) |

(b) Controlling $Fdr_{complete} \leq 5\%$ using lfsr statistic

| Method | S | K | Hit rate in % when data are | | Estimated error rate in % (95% CI) | | |
| --- | --- | --- | --- | --- | --- | --- | --- |
| | | | measured | missing | $\widehat{Fdr}_{complete}$ | $\widehat{Fdr}_{single}$ | $\widehat{Fsr}_{replicate}$ |
| XD | 1 | | 2.0 (1.9, 2.1) | 0.2 (0.2, 0.3) | 4.9 (3.8, 6.2) | 1.7 (1.3, 2.2) | 0.0 (0.0, 2.1) |
| XD | 2 | | 2.4 (2.3, 2.6) | 0.3 (0.3, 0.4) | 4.9 (3.8, 6.2) | 1.6 (1.3, 2.1) | 0.0 (0.0, 1.9) |
| mash | 158 | | 2.5 (2.4, 2.7) | 0.2 (0.2, 0.2) | 4.9 (3.8, 6.3) | 1.2 (0.8, 1.5) | 0.0 (0.0, 2.1) |
| ComposeMV | 1 | 15 | 8.9 (8.6, 9.2) | 1.6 (1.5, 1.7) | 5.0 (4.2, 5.9) | 1.5 (1.2, 1.8) | 1.7 (0.9, 3.2) |
| ComposeMV | 1 | 20 | 9.3 (8.9, 9.6) | 1.1 (1.0, 1.2) | 5.0 (4.2, 5.9) | 1.6 (1.3, 1.9) | 1.2 (0.5, 2.6) |
| ComposeMV | 1 | 30 | 8.4 (8.1, 8.7) | 0.9 (0.8, 0.9) | 5.0 (4.2, 5.9) | 1.7 (1.4, 2.0) | 1.0 (0.4, 2.5) |
| ComposeMV | 1 | 40 | 8.0 (7.8, 8.3) | 0.8 (0.7, 0.8) | 5.0 (4.2, 5.9) | 1.7 (1.4, 2.0) | 0.7 (0.2, 2.0) |
| ComposeMV | 2 | 15 | 8.2 (7.9, 8.5) | 1.6 (1.5, 1.8) | 5.0 (4.2, 6.0) | 2.3 (1.7, 2.8) | 1.2 (0.5, 2.7) |
| ComposeMV | 2 | 20 | 8.2 (7.9, 8.5) | 1.5 (1.4, 1.7) | 5.0 (4.2, 6.0) | 2.2 (1.7, 2.7) | 1.3 (0.6, 2.7) |
| ComposeMV | 2 | 30 | 7.8 (7.5, 8.1) | 1.1 (1.0, 1.2) | 5.0 (4.2, 5.9) | 2.1 (1.6, 2.5) | 1.0 (0.4, 2.6) |
| ComposeMV | 2 | 40 | 8.0 (7.7, 8.3) | 1.0 (1.0, 1.1) | 5.0 (4.2, 5.9) | 2.1 (1.6, 2.6) | 0.5 (0.1, 1.8) |

(c) Controlling lfsr $\leq 5\%$ using lfsr statistic

| Method | S | K | Hit rate in % when data are | | Estimated error rate in % (95% CI) | | |
| --- | --- | --- | --- | --- | --- | --- | --- |
| | | | measured | missing | $\widehat{Fdr}_{complete}$ | $\widehat{Fdr}_{single}$ | $\widehat{Fsr}_{replicate}$ |
| XD | 1 | | 8.4 (8.2, 8.5) | 1.1 (1.1, 1.2) | 50.9 (49.2, 52.5) | 18.6 (18.0, 19.4) | 1.7 (1.0, 2.9) |
| XD | 2 | | 7.9 (7.8, 8.1) | 1.4 (1.3, 1.5) | 44.8 (43.2, 46.5) | 15.5 (14.8, 16.2) | 1.9 (1.1, 3.1) |
| mash | 158 | | 5.4 (5.2, 5.6) | 1.1 (1.0, 1.2) | 6.7 (5.7, 7.8) | 2.7 (2.1, 3.4) | 1.5 (0.8, 2.8) |
| ComposeMV | 1 | 15 | 5.8 (5.7, 6.0) | 1.5 (1.5, 1.6) | 6.2 (5.3, 7.1) | 2.4 (2.1, 2.8) | 4.5 (3.3, 6.2) |
| ComposeMV | 1 | 20 | 5.5 (5.4, 5.7) | 0.9 (0.8, 0.9) | 5.8 (4.9, 6.7) | 2.5 (2.1, 2.9) | 2.3 (1.4, 3.7) |
| ComposeMV | 1 | 30 | 5.9 (5.7, 6.0) | 0.9 (0.8, 1.0) | 6.8 (5.9, 7.8) | 2.9 (2.5, 3.4) | 2.1 (1.3, 3.6) |
| ComposeMV | 1 | 40 | 5.8 (5.7, 6.0) | 0.8 (0.8, 0.9) | 6.6 (5.7, 7.6) | 2.9 (2.5, 3.4) | 3.0 (1.9, 4.6) |
| ComposeMV | 2 | 15 | 6.2 (6.0, 6.4) | 1.9 (1.8, 2.0) | 6.9 (6.0, 7.9) | 3.2 (2.6, 3.9) | 7.3 (5.6, 9.5) |
| ComposeMV | 2 | 20 | 6.2 (6.0, 6.4) | 1.8 (1.7, 1.9) | 7.2 (6.2, 8.2) | 3.7 (3.1, 4.4) | 3.5 (2.4, 5.0) |
| ComposeMV | 2 | 30 | 6.2 (6.0, 6.4) | 1.3 (1.2, 1.4) | 7.2 (6.2, 8.2) | 3.5 (2.9, 4.2) | 2.9 (1.9, 4.4) |
| ComposeMV | 2 | 40 | 6.1 (5.9, 6.3) | 1.2 (1.1, 1.2) | 6.9 (6.0, 7.9) | 3.5 (2.9, 4.1) | 3.5 (2.3, 5.3) |

lfsr, local false sign rate; mash, multivariate adaptive shrinkage; UV, univariate; XD, Extreme Deconvolution.

**Table 8. Comparison of cross-validated log likelihood across MV models on the IMPC data.** The row showing the model used in the main analyses in the paper, $\mathcal{M}_{\text{main}}$, is highlighted in bold. The largest CV log likelihood is under-lined. The results shown are the per-sample log likelihood $\mathcal{L}_{\text{CV}}^{(c)}$ of (34) averaged across folds $c = 1,\ldots,10$, along with $\pm 2$ SEM intervals.

| Method | S | K | CV Log Likelihood |
|--------|---|---|-------------------|
| XD | 1 | | −62.1 (−62.5, −61.7) |
| XD | 2 | | −60.0 (−60.7, −59.3) |
| mash | 158 | | −54.2 (−54.4, −53.9) |
| ComposeMV | 1 | 15 | −53.7 (−53.9, −53.5) |
| **ComposeMV** | **1** | **20** | **−53.4 (−53.6, −53.1)** |
| ComposeMV | 1 | 30 | −53.1 (−53.3, −52.8) |
| ComposeMV | 1 | 40 | −53.0 (−53.2, −52.7) |
| ComposeMV | 2 | 15 | −53.7 (−54.0, −53.4) |
| ComposeMV | 2 | 20 | −53.5 (−53.9, −53.2) |
| ComposeMV | 2 | 30 | −53.3 (−53.6, −53.0) |
| ComposeMV | 2 | 40 | −53.1 (−53.4, −52.9) |

CV, cross-validated; mash, multivariate adaptive shrinkage; SEM, standard error of the mean; XD, Extreme Deconvolution.

corresponds to a (gene, single nucleotide polymorphism) pair. These expression quantitative trait loci (eQTLs) are measured in 44 tissues. So, $N_{\text{tot}} = 16,069$ and $P = 44$, in contrast to $N_{\text{tot}} = 4,584$ and $P = 148$ for the IMPC data.

We analyse the eQTL data using the empirical Bayes cross-validation framework laid out in Methods–*Cross-validation and model averaging*. We follow the methods of [22] for data pre-processing and estimation of **R**. The size of our training folds on the eQTL data are 5,000 (in contrast to 2,000 for the IMPC data), but otherwise, parameter settings are the same. One important difference is that there are no missing data in the eQTL study. Table 9 shows the CV log likelihood comparison on the eQTL data set.

**Discussion of benchmarking results.** The main results we have presented our based upon the MV model (ComposeMV) with $S = 1$ and $K = 20$ (notation introduced in Results–

**Table 9. Comparison of cross-validated log likelihood across MV models on the eQTL data from Urbut and colleagues [22].** The largest CV log likelihood is underlined. The results shown are the per-sample log likelihood $\mathcal{L}_{\text{CV}}^{(c)}$ of (34) averaged across folds $c = 1,\ldots,10$, along with $\pm 2$ SEM intervals.

| Method | S | K | CV Log Likelihood |
|--------|---|---|-------------------|
| XD | 1 | | 25.75 (25.63, 25.87) |
| XD | 2 | | 28.21 (28.19, 28.23) |
| mash | 54 | | 35.44 (35.41, 35.48) |
| ComposeMV | 1 | 15 | 35.67 (35.60, 35.74) |
| **ComposeMV** | **1** | **20** | **35.68 (35.61, 35.74)** |
| ComposeMV | 1 | 30 | 35.68 (35.62, 35.74) |
| ComposeMV | 1 | 40 | 35.68 (35.62, 35.74) |
| ComposeMV | 2 | 15 | 36.21 (36.17, 36.26) |
| ComposeMV | 2 | 20 | 36.20 (36.16, 36.25) |
| ComposeMV | 2 | 30 | 36.21 (36.16, 36.25) |
| ComposeMV | 2 | 40 | 36.21 (36.16, 36.25) |

CV, cross-validated; eQTL, expression quantitative trait loci; mash, multivariate adaptive shrinkage; MV, multivariate; SEM, standard error of the mean; XD, Extreme Deconvolution.

*Multivariate model*) while controlling $\widehat{\mathrm{Fdr}}_{\mathrm{complete}} < 5\%$ as described in Methods–*Control of error rates*. We refer to this model as $\mathcal{M}_{\mathrm{main}}$ in this section. We now briefly discuss the results of benchmarking across 12 models under various means of error rate control (Tables 7 and 8).

Focusing on Table 7A, where we control $\widehat{\mathrm{Fdr}}_{\mathrm{complete}} < 5\%$, the hit rate on measured data for $\mathcal{M}_{\mathrm{main}}$ is the largest across all benchmarked models at 10.5% (10.2, 10.9). This hit rate is also optimal when compared to the other 2 considered methods of error rate control (Table 7B and 7C). It's worth noting, when comparing hit rates to Table 7C, that the monitored error rates in Table 7C are generally higher than in Table 7A and 7B, attributable to using a different method for error rate control, nominally controlling lfsr<5%.

In terms of optimal hit rates when data are missing, we see that other models (e.g., ComposeMV with $S = 2$ and $K = 15$) performed slightly better with 2.0% (1.8, 2.1) compared to 1.3% (1.2, 1.4) for $\mathcal{M}_{\mathrm{main}}$, but this comes with a higher estimated error rate $\widehat{\mathrm{Fsr}}_{\mathrm{replicate}}$ compared to $\mathcal{M}_{\mathrm{main}}$.

Turning to the cross-validated likelihood comparison in Table 8, we see that the mean per-sample cross-validated log likelihood for $\mathcal{M}_{\mathrm{main}}$ is $\widehat{\mathcal{L}}_{\mathrm{CV}} = -53.4$ (−53.6, −53.1). This is marginally improved upon by ComposeMV with $S = 2$ and $K = 15$ having $\widehat{\mathcal{L}}_{\mathrm{CV}} = -53.0$ (−53.2, −52.7). $\mathcal{M}_{\mathrm{main}}$, and ComposeMV generally, compare favourably with existing methods, performing somewhat better than mash which has $\widehat{\mathcal{L}}_{\mathrm{CV}} = -54.2$ (−54.4, −53.9) and considerably better than XD with $S = 2$, which has $\widehat{\mathcal{L}}_{\mathrm{CV}} = -60.0$ (−60.7, −59.3).

For the benchmarking on an additional eQTL data set, shown in Table 9, a similar pattern emerges—the current paper's MV model, labelled ComposeMV, performs somewhat better in terms of CV likelihood compared to mash, while performing substantially better than XD. Interestingly, for the eQTL data benchmarking, the ComposeMV models with $S = 2$ perform best of those in the table, which suggests that having a mixture of multiple learned covariance matrices (in addition to the multiscale ladder of the $\omega_m$) may be particularly useful in certain contexts.

## Supporting information

**S1 Note. Univariate model details.** This note details the technical aspects of the UV model introduced in Results–*Univariate model*.
(PDF)

**S2 Note. 2. EM algorithm.** This note contains technical details of the EM algorithm introduced in Methods–*EM algorithm*.
(PDF)

**S1 Fig. The IMPC adult and embryonic phenotype pipeline.** Scientific purpose, experimental design, and detailed description for each procedure are available at www.mousephenotype.org/impress/pipelines. Each phenotype within each procedure is also described in detail. Note that the terminology *parameters* is used there to refer to what we call *phenotypes* in this paper. We prefer to use *phenotypes* to avoid any terminological ambiguity with the use of parameters in statistical inference.
(TIF)

**S2 Fig. Heatmap of scaled *z*-statistics illustrating the quality control filter applied to UV results.** KO lines are ordered horizontally by time within centre. Longitudinal trends within a phenotyping centre can be indicative of experimental artefacts not captured by the UV model. In such cases, outlined with red rectangles, the data from centre–procedure pairs are omitted

from downstream MV analysis. The data and code used to generate this figure are available at [13,14]. KO, knockout; MV, multivariate; UV, univariate.
(TIF)

**S3 Fig. Scatterplot of $\widetilde{z}_{pg}$ for MV against UV models for gene-phenotype pairs at which data are available.** The axes extend to $[-3, 3]$ while the counts apply to all data, including those beyond the scale of the plot. The data and code used to generate this figure are available at [13,14]. MV, multivariate; UV, univariate.
(TIF)

**S4 Fig. Power and % missing data by phenotype.** The top panel shows the % missing data for each phenotype. The lower panel displays the phenotype-specific hit rate (i.e., proportion of lines that are significantly perturbed), for the UV method, and for the MV method stratified according to whether data are missing or observed. The data and code used to generate this figure are available at [13,14]. MV, multivariate; UV, univariate.
(TIF)

**S5 Fig. Replicability heatmap comparing results across phenotyping centres.** The heatmap shows scaled $z$-statistics, $\widetilde{z}$, for reference lines under the UV and MV models. Significant perturbations ($|\widetilde{z}| > 1$) are marked with a cross. White squares represent missing data under the UV model. Seven KO lines are shown (labelled top) measured independently in several laboratories (labelled bottom) and analysed using the UV and MV models (labelled third row from top). Each row corresponds to a phenotype (labelled right), grouped by procedure (labelled left). The data and code used to generate this figure are available at [13,14]. KO, knockout; MV, multivariate; UV, univariate.
(TIF)

**S6 Fig. Heterozygote/homozygote concordance scatterplot of scaled $z$–statistics, $\widetilde{z}_{pg}$.** Each point corresponds to the $\widetilde{z}_{pg}$ of the heterozygote and homozygote KO lines of a particular gene. Counts (%) for each significance combination are superimposed; while the axes extend to $[-3, 3]$, the counts apply to all data, including those beyond the plot's scale. An Fsr estimate $\widehat{\mathrm{Fsr}}_{\mathrm{replicate}}$ (95% CI) based on the level of discordance is shown at the top of the panel. The data and code used to generate this figure are available at [13,14]. Fsr, false sign rate; KO, knockout.
(TIF)

**S7 Fig. Co-enrichment of GO terms (left) with IMPC phenotypes (bottom) <u>for hits called by UV model</u>.** Statistically significant co-enrichment between GO terms and IMPC phenotypes is denoted by bold outlined squares (controlling family-wise error rate <5% for each phenotype). The colour of the square indicates the percentage of significantly perturbing KO genes at the GO term that change the phenotype in the positive direction (see scale bar at top). IMPC phenotypes are clustered by GO term pattern along the horizontal axis, while BP GO terms are clustered vertically by phenotype pattern. Phenotype labels are coloured according to procedure as per legend at bottom left. A subset of GO terms, labelled by row (a-h) at right, are examined in more detail in Fig 7. For legibility, we only plot IMPC phenotypes and GO terms that have at least 3 instances of significant co-enrichment. The data and code used to generate this figure are available at [13,14]. BP, Biological Process; GO, Gene Ontology; IMPC, International Mouse Phenotyping Consortium; KO, knockout.
(TIF)

**S8 Fig. Cumulative proportion of correlation structure in $\widehat{\Sigma}$ explained by eigenvectors $Q$ in (31).** The dotted line indicates that over 75% of the correlation is explained by 20 eigenvectors.

The data and code used to generate this figure are available at [13,14].
(TIF)

**S9 Fig. Scatterplots of $\widetilde{z}_{pg} := z_{pg}/\tau_{pc}$ examining concordance of MV analysis on masked data (LOO-MV) with the UV model.** We plot LOO-MV results (inferring perturbations on masked data) against results for the UV model applied to the unmasked data; see Methods–*Predicting masked data*. An Fsr estimate $\widehat{\text{Fsr}}_{\text{replicate}}$ (95% CI) based on the level of discordance is shown at the top of the panel. The data and code used to generate this figure are available at [13,14]. Fsr, false sign rate; LOO-MV, leave-one-procedure-out MV; MV, multivariate; UV, univariate.
(TIF)

**S10 Fig. Comparison of cross-validated (CV) log likelihood between randomly initialised and sample-covariance initialised fits for 10 CV folds.** The randomly initialised fits perform systematically worse in terms of CV likelihood, supportive of using a supervised initialisation to mitigate the non-convexity of the optimisation. The data and code used to generate this figure are available at [13,14].
(TIF)

**S11 Fig. Sensitivity analysis of factor loadings.** (a) The varimax-rotated loadings for $\widehat{\mathbf{\Sigma}}_{\text{pooled}}$, the Bayesian model averaged covariance matrix across all cross-validation folds. (b) The varimax-rotated loadings for the fold $c'$ covariance matrix $\widehat{\mathbf{\Sigma}}^{(c')}$, which is chosen to maximise the symmetrized KL divergence between $\text{N}(0, \widehat{\mathbf{\Sigma}}_{\text{pooled}})$ and $\text{N}(0, \widehat{\mathbf{\Sigma}}^{(c)})$ across folds $c$. The 2 loadings plots are qualitatively similar, though there are some small discrepancies. The data and code used to generate this figure are available at [13,14].
(TIF)

## Author Contributions

**Conceptualization:** George Nicholson, Habib Ganjgahi, Steve D. M. Brown, Ann-Marie Mallon, Chris Holmes.

**Data curation:** George Nicholson, Hugh Morgan.

**Formal analysis:** George Nicholson, Hugh Morgan, Habib Ganjgahi.

**Investigation:** George Nicholson, Hugh Morgan.

**Methodology:** George Nicholson, Hugh Morgan, Habib Ganjgahi, Chris Holmes.

**Visualization:** George Nicholson.

**Writing – original draft:** George Nicholson, Steve D. M. Brown, Chris Holmes.

**Writing – review & editing:** George Nicholson, Steve D. M. Brown, Ann-Marie Mallon, Chris Holmes.

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
