## [Editor Report · Decision Letter 0]

17 Jan 2020

Dear Dr Nicholson, 

Thank you for submitting your manuscript entitled "Illuminating the mammalian genome with multivariate phenotype analysis" for consideration as a Research Article by PLOS Biology.

Your manuscript has now been evaluated by the PLOS Biology editorial staff, as well as by an academic editor with relevant expertise, and I'm writing to let you know that we would like to send your submission out for external peer review. IMPORTANT: We have decided that your paper would be best reviewed as a Methods and Resources paper; no re-formatting is required, but please can you select the article type "Methods and Resources" when you upload your additional metadata (see next paragraph)?

Please re-submit your manuscript within two working days, i.e. by Jan 21 2020 11:59PM.

Kind regards,

Roli Roberts

Senior Editor

PLOS Biology

---

## [Decision Letter · Decision Letter 1]

24 Mar 2020

Dear Dr Nicholson,

Thank you very much for submitting your manuscript "Illuminating the mammalian genome with multivariate phenotype analysis" for consideration as a Methods and Resources at PLOS Biology. Your manuscript has been evaluated by the PLOS Biology editors, an Academic Editor with relevant expertise, and by two independent reviewers. We recruited a third reviewer, but because of the Covid-19 crisis this reviewer has asked if they can send their comments on later (see below).

You’ll see that both reviewers #1 and #2 are broadly positive about your study, but have various suggestions (mostly presentational) how to improve it. Reviewer #1's requests are almost all presentational, but they’re quite substantial and, for a broad-readership journal like PLOS Biology, should be considered essential. Reviewer #2 would like you to include an explicit comparison between your approach and that of Urbut et al. (https://www.nature.com/articles/s41588-018-0268-8), to showcase its generality by including an additional worked example, and to do some sensitivity analyses. The Academic Editor has asked me to stress that "the paper would not be accepted unless a systematic comparison with Urbut as well as broader biological inference can be added."

IMPORTANT: We recruited a third reviewer who is struggling to finalise their comments during preparations for Covid-19 shut-down. They aim to submit their comments by the end of this week, and told us that while they agree in principle with the decision to revise, "there are additional analyses / checks to be done to make this paper more balanced." I will transmit this reviewers' comments to you as soon as I receive them; I hope you understand our reasons for doing this.

In light of the reviews (below), we will not be able to accept the current version of the manuscript, but we would welcome re-submission of a much-revised version that takes into account the reviewers' comments. We cannot make any decision about publication until we have seen the revised manuscript and your response to the reviewers' comments. Your revised manuscript is also likely to be sent for further evaluation by the reviewers.

We expect to receive your revised manuscript within 2 months. 

**IMPORTANT - SUBMITTING YOUR REVISION**

*Re-submission Checklist*

*Published Peer Review*

*PLOS Data Policy*

*Blot and Gel Data Policy*

Sincerely,

Roli Roberts

Senior Editor

PLOS Biology

REVIEWERS' COMMENTS:

Reviewer #1:

[identifies himself as Richard Mott]

This MS describes a sophisticated multivariate analysis of over 6,000 IMPC gene knockouts measured on 148 phenotypes. The analysis aims to call correctly the phenotypic effect for each gene on each phenotype at a controlled false discovery rate, and to impute missing results where possible. This dataset is difficult to work with because of its internal structure with multiple levels of correlation.

The analysis methodology is a hybrid of Bayesian and Frequentist steps, which makes it rather hard to follow in places. While I don't have any substantive concerns (other than those noted below) one is left with the feeling that the pipeline is over-complicated. I think this is mainly because the MV analysis has been bolted onto the outputs of the UV analysis. Nonetheless, the results are encouraging, in that a larger fraction of positive phenotype calls are made using multivariate analysis compared to univariate analysis, whilst controlling the FDR. As far as I can tell, without redoing the analysis, the results are correct, and the models fitted are sensible. This alone is a very significant achievement.

There are many important methodological details and caveats which are buried in the methods, and which makes the MS hard to follow. The MS has a large number of figures (particularly heatmaps) which are not always very informative. The authors should review carefully whether all these figures are necessary or could be moved to the supplement. Some simple pie charts, or similar, giving the fractions of positive phenotype calls, sex effects, how much the imputation increased the fraction of positive calls etc, would be helpful additions.

This is not meant to detract from the considerable achievement of a generally excellent statistical analysis, but I wonder what a biologist will make of the findings. For instance, there is no attempt to use the increased phenotype call rate to make inferences about which gene sets are involved in phenotypes, and if these sets are related to known gene networks, or possibly physically clustered on the genome. 

Finally, what is the status of these new phenotype calls - will they replace the existing IMPC annotations? How do the positive gene annotations generated in the current MS differ from those reported elsewhere? What is the desired end point: is this a methods paper, or the definitive IMPC analysis? 

Technical points requiring clarification

1. Although the terminology is used in the mouse phenotyping community, it is confusing to use "annotate" to mean "call a gene positive for a phenotype" - in the wider community "annotate" means to test a gene for a phenotype and to call the result either positive or negative (thereby distinguishing it from missing data). This specialised use of the word "annotation" should either be prominently flagged or replaced by e.g. "positive annotation".

2. It is unclear from the methods how centre effects are modelled. Are they fixed or random effects, or are they ignored at the UV stage and investigated later? Are they controlled for, eg in the selection of control animals? Similarly where a gene is tested at multiple centres, are the data treated like different genes and then compared later on?

3. Sex effects are removed at the UV stage and treated essentially as nuisance variables. But we think from earlier analyses of IMPC (ref 5 in the current MS) that about 12% of genes have sex-specific effects (ie an interaction), which is not handled by a simple sex effect that affects cases and controls equally, and which is what was presumably fitted in the MS. So far as I can see by the MV stage of the analysis it is impossible to see where sex has any effect, because the UV step removes these effects. At the very least, this requires some discussion.

4. A major analysis hurdle for the IMPC data is the correct selection of controls. This happens upstream of the other analyses in the current MS (I think) so it is impossible to tell if the selection method used in the current study is correct (but equally, no reason to suppose it is problematic either). There has already been some work on this problem (ref 9 in the MS). It is not clear if the method of choosing controls in the current analysis differs from that done previously and whether the phenotype calling is affected. In the Methods in the UV analysis states that it is "….fitted only to data from KO line g accompanied by data from the entire rolling baseline of WT animals…". Does this mean all controls are compared to a given KO? But later the Methods describes Negative Control Data as "We generate negative control data and use them to enhance inference under the univariate (UV) and multivariate (MV) models. Negative-control lines are generated by randomly selecting groups of WT animals so as to match the experimental design characteristics of a particular true KO line…" This is confusing: please clarify what is going on. It appears in the Calibration section of the Methods as if negative controls were treated as if they were positives for calibration of thresholds. This is different from defining a set of controls for each gene. It is essential to describe and justify the control selection strategy used (see also comments about centre effects, above). 

Reviewer #2:

In this paper, the authors develop and apply a statistical framework for the joint analysis of high-dimensional phenotype response studies. Accurate statistical methods for the analysis of response data are very much needed, and the paper describes an application to a major data resource as part of the international mouse phenotyping consortium. The paper is well written and of broad interest. 

The paper is a combination of a new method and an application, although as presented the method is at the center. My main concern is the methodological contribution and the conceptual novelty of what is presented. As correctly referenced and stated, there have been alternative frameworks proposed to essentially perform the same type of analysis. While previous methods have been applied to somewhat different questions and data, e.g. multi-tissue expression QTL mapping, the statistical principles are clearly transferrable and it is not clear how the presented method relates and compares. 

Major comments:

1. Relationship to cited prior work.

The approach proposed by Urbut et al. [1] is quite similar to what is presented here, yet a comparison and even a discussion to relate the approach is missing. Given that this method could in principle also be applied to this study, a comparison would seem necessary. This may impact the relative weight of the methodology that is presented versus the application to the specific dataset, which however is not a problem per se. 

2. Scope of the paper and the method. 

The title and abstract are framed very general, yet the application that is considered is specific. Depending on how the authors decide to address the previous comment, it may be useful to consider additional application to additional data to demonstrate broad applicability. 

3. Downstream analysis of Sigma. 

The factor analysis downstream modelling is interesting, as is the comparison of direct phenotypic correlation versus the estimated effected size covariance structures. These analyses are based on potentially noisy ML-based estimates of Sigma, and it is not clear to me if and how the uncertainty in these estimates is accounted for in these analyses. At the very least some sort of sensitivity analysis should be conducted to demonstrate robustness. 

References:

[1] Urbut, Sarah M., et al. "Flexible statistical methods for estimating and testing effects in genomic studies with multiple conditions." Nature genetics 51.1 (2019): 187-195.

---

## [Decision Letter · Decision Letter 2]

20 May 2022

Dear George,

Thank you for your patience while we considered your revised manuscript "Illuminating the mammalian genome with multivariate phenotype analysis" for publication as a Methods and Resources at PLOS Biology. This revised version of your manuscript has been evaluated by the PLOS Biology editors, the Academic Editor and one of the original reviewers. The Academic Editor has kindly assessed your responses to the other two reviewers.

Based on the review and our Academic Editor's assessment of your revision, we are likely to accept this manuscript for publication, provided you satisfactorily address the remaining points raised by reviewers #1 and the following data and other policy-related requests.

IMPORTANT: Please attend to the following:

a) Please address the remaining points raised by reviewer #1.

b) Please provide a more informative and declarative Title. We suggest something like the following: "Multivariate phenotype analysis enables genome-wide inference of mammalian gene function"

c) Please provide a blurb according to the instructions in the submission form.

d) My understanding is that your GitHub deposition contains the data and code required to generate all of data-containing Figure panels, namely Figs 1AB, 2AB, 3AB, 4, 5ABC, 6ABCD, 7, 8AB, S2, S3, S4, S5, S6, S7, S8, S9, S10AB, S11. Can you confirm that this is the case?

e) As GitHub can be changed at any time, please can you also generate a deposited immutable version in a repository such as Zenodo?

f) Please then cite the GitHub and Zenodo URLs as the location of the data in each relevant main and supplementary Figure legend (e.g. "The data and code needed to generate this Figure can be found in XYZ").

We expect to receive your revised manuscript within two weeks. 

*Published Peer Review History*

*Press*

Sincerely,

Roli

Senior Editor,

rroberts@plos.org,

PLOS Biology

DATA POLICY:

[Figs….]

DATA NOT SHOWN?

REVIEWER'S COMMENTS:

Reviewer #1:

[identifies himself as Richard Mott]

The authors have satisfactorily addressed my questions in the revision. 

I have a few suggestions for clarity and readability which I would hope the authors will consider, but these are not impediments to publication. 

It's a very nice study.

1. The Methods contains some important results on robustness of the results, in particular the cross-validations to show that missing data are imputed satisfactorily. I suggest you move at least a summary of these analyses into the main Results, since you raise the problem of missing data in the introduction but don't directly address it in the main Results in the current revision. Line 230 near Fig 2 might be a good place to introduce this.

2. Explain clearly in the Results what the categorisation of calls as -1, 0 , 1 represents (eg legend to Table 1) - this seems to be buried in the Methods 

3. Explain more clearly what the factor analysis means from a biological perspective. I took it to mean there are clusters of phenotypes that tend to get called in the same way (is that correct?). In the abstract they are described as "sparse factors" which has an obscure technical meaning, losing clarity.

4. In Fig 1, it might be helpful to show both the UV and MV estimators and their confidence bands, for illustrative purposes. In general, are the MV estimates more precise than the UV? (sensu smaller standard errors)

5. In the abstract there is a mention of "7.5-fold increase in statistical power". It's not clear that this really means "There are 4,256 (1.4% of 302,997 observed-data measurements) hits called by the UV model, compared to 31,843 (10.5%) hits in the observed-data results of the MV model". Suggest you expand the text in the abstract.

---

## [Editor Report · Decision Letter 3]

22 Jun 2022

Dear George,

Thank you for the submission of your revised Methods and Resources paper, "Multivariate phenotype analysis enables genome-wide inference of mammalian gene function" for publication in PLOS Biology. On behalf of my colleagues and the Academic Editor, Nicole Soranzo, I'm pleased to say that we can in principle accept your manuscript for publication, provided you address any remaining formatting and reporting issues. These will be detailed in an email you should receive within 2-3 business days from our colleagues in the journal operations team; no action is required from you until then. Please note that we will not be able to formally accept your manuscript and schedule it for publication until you have completed any requested changes.

Sincerely,

Roli

Senior Editor

PLOS Biology

rroberts@plos.org